

# Reactive bromine in the low troposphere of Antarctica. Estimations at two research sites.

Cristina Prados-Roman[1], Laura Gómez-Martín[1,2], Olga Puentedura[1], Mónica Navarro-Comas[1], Javier Iglesias[1], José Ramón de Mingo[3], Manuel Pérez[*], Héctor Ochoa[4], María Elena Barlasina[5], Gerardo Carbajal[5,6] and Margarita Yela[1]

[1]Atmospheric Research and Instrumentation Branch, National Institute for Aerospace Technology (INTA), Madrid, 28850, Spain.

[2]Groupe de Spectrométrie Moléculaire et Atmosphérique, URM CNRS 7331, UFR Sciences Exactes et Naturelles, Moulin de la Housse, BP 1039, 51687 Reims Cedex 2, France.

[3]Space Sensors Engineering, National Institute for Aerospace Technology (INTA), Madrid, 28850, Spain.

[4]National Antarctic Direction (DNA)/Argentinian Antarctic Institute (IAA), 25 de Mayo 1143, San Martín Provincia de Buenos Aires, Argentina.

[5]National Meteorological Service (SMN), Atmospheric Watch and Geophysical (GIDyC–VAYGEO), Buenos Aires, Argentina.

[6]Pontificia Universidad Católica Argentina, PEPACG, Facultad de Ingeniería y Ciencias Agrarias, Buenos Aires, Argentina.

[*]formerly at the Atmospheric Research and Instrumentation Branch, National Institute for Aerospace Technology (INTA), Madrid, 28850, Spain.

*Correspondence to*: Cristina Prados-Roman (pradosrc@inta.es)

**Abstract.**

For decades, reactive halogen species (RHS) have been subject of detailed scientific research due to their influence on the oxidizing capacity of the atmosphere and on the climate. From the RHS, those containing bromine are of particular interest in the polar troposphere as a result of their link to ozone depletion events (ODEs) and to the perturbation of the cycle of e.g. the toxic mercury. Given its remoteness and related limited accessibility compared to the Arctic region, the RHS in the Antarctic troposphere are still poorly characterized. This work presents ground-based observations of tropospheric BrO from two different Antarctic locations: Marambio (64º 13' S, 56º 37' W) and Belgrano II (77º 52' S, 34º 37' W) during the sunlit period of 2015. By means of MAX-DOAS (Multi-axis Differential Optical Absorption Spectroscopy) measurements of BrO performed from the two research sites, the seasonal variation of this reactive trace gas is described along with its vertical and geographical distribution in the Antarctic environment. Results show an overall vertical profile of BrO mixing ratio decreasing with altitude, with a median value of 1.6 pmol mol$^{-1}$ in the lowest layers of the troposphere and undetectable values above 2 km at both sites. Additionally, observations show that the polar sunrise triggers a heterogeneous increase of bromine content in the Antarctic troposphere yielding a maximum BrO at Marambio (26 pmol mol$^{-1}$), amounting threefold the values observed at Belgrano at dawn. Data presented herein are combined with previous studies and ancillary data to update and expand our knowledge of the geographical and vertical distribution of BrO in the Antarctic troposphere, revealing Marambio as one of the locations with highest BrO reported so far in Antarctica. Furthermore, the observations gathered during 2015 serve as a proxy to investigate the budget of reactive bromine (BrO$_x$ = Br + BrO) and the bromine-mediated ozone loss rate in the Antarctic troposphere.

## 1 Introduction

The importance of the halogens (X= Cl, Br, I) in atmospheric chemistry and climate became clear decades ago after observations of these substances were made in the stratosphere and also in the troposphere (e.g., Molina and Rowland, 1974; Farman et al., 1985; Barrie et al., 1988; Oltmans et al., 1989; Fan and Jacob, 1992; Hausmann and Platt, 1994; Solomon, 1999). Indeed, reactive halogen compounds (RHS) are of special interest in the troposphere for limiting the lifetime of species such



ozone ($O_3$), mercury (Hg), dimethyl sulfide (DMS) and organic compounds; for affecting the partitioning of $NO_x$ ($NO + NO_2$) and $HO_x$ ($OH + HO_2$) and, in the case of iodine, for participating in new aerosol formation. As such, the presence and the impact of the tropospheric halogen chemistry have been subject of numerous studies with focus on remote regions and on environments under anthropogenic influence (e.g., Simpson et al., 2015 and references therein). Particular attention has been

paid by the scientific community to the role of halogens in the polar regions. Although not unique to these regions, it is in the polar areas where bromine becomes particularly relevant directing the oxidizing capacity of the atmosphere during springtime and causing ozone and mercury depletion events (ODEs and AMDEs, respectively). For details on sources, sinks and historical background, the reader is kindly referred to the compendium works of e.g. Simpson et al. (2007), Steffen et al. (2008), Ariya et al. (2015) and Simpson et al. (2015) and references therein.

Briefly, while the presence of reactive bromine in the global pristine troposphere is primarily due to the photolysis and oxidation of very short-lived bromocarbons emitted from the oceans (e.g., Carpenter and Reimann, 2014), in the polar regions its dominant source is of inorganic origin and is linked to heterogeneous chemistry. Through experimental and modelling studies, it is known that a set of heterogeneous reactions based on acidic substrates comprising hypobromus acid (HOBr) and brominde ($Br^-$) take place in e.g. sea ice, open leads, brine, frost flowers, snowpacks or sea-salt aerosols (e.g., Fan and Jacob,

1992; Vogt et al., 1996; Platt and Lehrer, 1997; Abbatt et al., 2012; Pratt et al., 2013, Toyota et al., 2014; Simpson et al., 2015, Thompson et al., 2015 and 2017, Custard et al., 2017; Wang and Pratt, 2017). These reactions are summarized as:

$$(HOBr)aq + (H^+)aq + (Br^-)aq \leftrightarrow (Br_2)aq + H_2O \quad (R1)$$

or

$$(HOBr)aq + (H^+)aq + (Cl^-)aq \leftrightarrow (BrCl)aq + H_2O \quad (R2)$$

$$(BrCl)aq + (Br^-)aq \leftrightarrow (Br_2Cl^-)aq \quad (R3)$$

$$(Br_2Cl^-)aq \leftrightarrow (Br_2)aq + (Cl^-)aq \quad (R4)$$

yielding the possibility that molecular bromine ($Br_2$) transforms from the aqueous (aq) to the gas phase. When this is followed by the photolysis of $Br_2$ into two bromine atoms, an autocatalytic release of bromine is triggered resulting in an exponential buildup of reactive bromine $BrO_x$ ($Br + BrO$) in the troposphere and the so-called "Bromine Explosions" events (e.g., Fan and

Jacob, 1992; Platt and Lehrer, 1997; Wennberg, 1999; Simpson et al., 2015). These events were firstly observed in the Arctic region by correlating the detection of filterable bromine with ODEs (e.g., Barrie et al., 1988) and, later on, by observing very high amount of BrO (tens of pmol mol$^{-1}$) in the boundary layer just after the polar sunrise (e.g., Hausmann and Platt, 1994; Tuckermann et al., 1997). Since then, several studies have tried to determine the chemical sources, sinks and pathways of these compounds (e.g., Simpson et al., 2007 and 2015). In particular, the main BrO source reactions involve:

$$Br_2 \overset{\lambda}{\to} 2Br \quad (R5)$$

$$BrCl \overset{\lambda}{\to} Br + Cl \quad (R6)$$

$$Br + O_3 \to BrO + O_2 \quad (R7)$$

In pristine environments (i.e., very low nitrogen oxide), along with photodissociation (in polar spring $J_{BrO} \sim 3 \cdot 10^{-2}$ s$^{-1}$, e.g., Thompson et al., 2015), the BrO sink reactions associated with the catalytic ODEs are:

$$BrO + BrO \to 2Br \quad (R8a)$$

$$\to Br_2 \quad (R8b)$$

$$BrO + ClO \to BrCl \quad (R9a)$$

$$\to Br + Cl \quad (R9b)$$

$$BrO + HO_2 \to HOBr \quad (R10)$$

$$BrO + OH \to Br + HO_2 \quad (R11)$$

where R10 represents the main channel for the above mentioned bromine explosions causing ODEs (e.g., Bottenheim et al., 1986; Barrie et al., 1988; Oltmans et al., 1989; Platt and Hönninger, 2003; Simpson al., 2007), where the ozone loss rate is



limited by the BrO self and cross reactions (R8 and R9) and estimated as (e.g, Hausmann and Platt, 1994; Le Bras and Platt, 1995; Platt and Jenssen, 1995; Platt and Lehrer, 1997):

$$-\frac{d[O_3]}{dt} = 2 \left( k_{BrO+BrO} [BrO]^2 + k_{BrO+ClO} [BrO][ClO] \right) \quad (1)$$

Overall, the attempts from the scientific community to estimate the presence of $BrO_x$ in the Antarctic troposphere initiated 20

years ago with ground-based DOAS measurements of BrO from Arrival Heights (77.8º S, 166.7º E), observations compatible with the presence of 30 pmol mol[-1] in the first 2 km of the troposphere (Kreher et al., 1997). Due to the complexity of performing measurements in such a hostile and remote environment, very few ground-based scientific works have followed that study (summarised in Table 1). In addition to the sparse ground-based measurements, the presence of tropospheric BrO in the Antarctic region has also been addressed through satellite observations (e.g., Wagner and Platt, 1998; Wagner et al.,

2001; Richter et al., 2002; Theys et al., 2011), ship-borne measurements (e.g., Wagner et al., 2007) and, more recently, by airborne DOAS measurements (e.g., Hüneke et al., 2017). In spite of the elapsed years and the efforts of the scientific community, compared to its northern counterpart, the current characterization of $BrO_x$ in the Antarctic troposphere is very poor given the very scarce geographical coverage available with vertical information. Moreover, most of the observations are campaign-based in random years and, hence, the time coverage is also quite limited (see Table 1). The present work aims at

improving this geographical, vertical and time coverage by adding two Antarctic sites to those few observing BrO in the Antarctic troposphere. These observations were made by endurable, stable and sensitive DOAS instrumentation built specifically for long-term measurements in hostile environments. Particularly, herein we present the observations performed during 2015 from two stations. The measurement sites and methodologies are introduced in Sect. 2. Section 3 puts forward the results obtained in terms of time series of BrO along with time series of the aerosol optical thickness, near surface $O_3$ and

meteorology parameters. Then it deepens into the details of the vertical information gained after these observations and assesses the budget and distribution of inorganic reactive bromine in the troposphere of Antarctica. Section 4 summarizes the work.

## 2 Observations from two Antarctic stations

During 2015, ground-based spectroscopic measurements were performed from two Antarctic research stations: Marambio and

Belgrano II. Details of the measurement sites and methods are provided below, along with ancillary observations.

### 2.1 Measurement Sites

In 2010, in collaboration with the National Antarctic Direction of Argentina / Argentinian Antarctic Institute (DNA/IAA), INTA deployed a MAX-DOAS (Multi-axis Differential Optical Absorption Spectroscopy, e.g., Platt and Stutz, 2008) instrument at the research base of Belgrano II (77º 52' S, 34º 37' W; 256 m a.s.l,), at the southern end of the Weddell Sea

(from now on referred to as "Belgrano"). Later on, in 2015, similar instrumentation was installed in the site of Marambio (64º 13' S, 56º 37' W; 198 m a.s.l.), located on Seymour Island (a small island just east of James Ross Island), in the northern tip of the Antarctic Peninsula. Since then, MAX-DOAS observations have been kept remotely. In 2016, both DOAS stations were accepted as part of the NDACC (Network for the Detection of Atmospheric Composition Change, http://www.ndsc.ncep.noaa.gov/), aiming at long-term atmospheric observations (e.g., De Mazière et al., 2017). Note that,

given their location around the Weddell Sea, long-term trace gas observations from these Antarctic sites provide a great opportunity for investigating the troposphere-sea ice interactions in two different scenarios: a station surrounded by seasonal sea ice (Marambio) and another where the perennial (edged) sea ice dominates (Belgrano). Figure 1 shows the locations where INTA has instrumentation deployed in Antarctica and the sea ice concentration (Spreen et al., 2008) surrounding these stations by the end of the austral summer and by mid-winter of the referred year 2015, which was the first year that both instruments

operated simultaneously.



## 2.2 Measuring Method

The spectral measurement technique used for the observations presented in this work was MAX-DOAS, gathering UV/VIS scattered skylight in the sunlit atmosphere. Through this technique, tropospheric vertical profiles of aerosols extinction coefficients (AEC) and BrO volume mixing ratios (vmr) can be inferred. Specific details of the instruments deployment and 5 the spectral analysis and inversion scheme are provided in the following.

### 2.2.1 MAX-DOAS instruments

Although in few occasions tropospheric BrO has been measured in remote regions with in situ techniques (e.g., chemical ionization mass spectrometry, Liao et al., 2011), the operational activities in remote and hostile environments renders the DOAS (Differential Optical Absorption Spectroscopy) technique as a very suitable approach given its sensitivity, versatility 10 and instrumental endurance (e.g., Platt and Stutz, 2008). Either with active set-ups (e.g., Long path-DOAS) or with passive ones (e.g., zenith-DOAS, MAX-DOAS, satellite observations), the DOAS technique has been used broadly to research the troposphere in remote environments (e.g., Wagner and Platt, 1998; Bobrowsky et al. 2003; Wagner et al., 2007; Saiz-Lopez et al., 2007a,b; Puentedura, et al. 2012, Prados-Roman et al., 2015; Peterson et al., 2017). In particular, the MAX-DOAS instrumental set-up referred to in this work consists in a telescope scanning the atmosphere at different elevation angles 15 inferring with it vertically-resolved information of the status of the atmosphere regarding aerosols and trace gases (e.g., Hönninger et al., 2004; Wagner et al., 2004). This is indeed an advantage that the MAX-DOAS configuration offers over the standard set-up of the e.g. long path-DOAS and also over in situ instruments (e.g., chemical ionization mass spectrometry, CIMS), whose information is commonly limited to the instrument's altitude. Also, the MAX-DOAS specific ability to characterize the low troposphere overcomes the often limited sensitivity of the satellite observations to the planetary boundary 20 layer.

The hardware and software of the MAX-DOAS instruments referred to in this work were developed by INTA. For decades, the group has been investigating the atmosphere from different sites of the world by means of the DOAS technique, particularly from polar regions (e.g., Gil et al., 1996; Gil et al., 2008; Yela et al., 2017). The two MAX-DOAS instruments referred to in this study consist of an outdoor unit with a temperate pointing system developed and built at INTA (Figure 2), comprising a 25 stepper motor and a telescope with an 8 cm focal length fused silica lens yielding a field of view of 1°. The sunlight is focused in a quartz fibre bundle which is directed into the indoor unit comprising a temperature-stabilised Czerny-Turner monochromator and a CCD camera fully developed by INTA based on an HAMAMATSU S7031-1008 sensor, kept at -40 °C ± 0.05 °C with a temperature control developed and built at INTA. Both instruments operate in off-axis mode scanning the atmosphere from the horizon to the zenith every 15 minutes while the solar zenith angle (SZA) is lower than 85°. For SZA 30 higher than 85°, the telescopes are fixed at zenith position and during the polar night no measurements are performed (i.e., April-August in Marambio and March-September in Belgrano). Further details of the MAX-DOAS instruments installed at Antarctica are provided in Table 2.

### 2.2.2 Spectral analysis and vertical profile inversion

The spectral analysis of the DOAS observations shown in this work was performed with the INTA's software LANA (e.g., 35 Gil et al., 2008; Peter et al., 2017). The retrieval of BrO was centered in the 335-358 nm spectral range, including the absorption cross-sections of BrO (Fleischmann et al., 2004), $O_4$ (Thalman and Volkamer, 2013), $CH_2O$ (Meller and Moortgat, 2000), OClO (Kromminga et al., 2003), $NO_2$ (Vandaele et al., 1998), $O_3$ (Bogumil et al., 2003) and of a pseudo-Ring spectra (Chance and Spurr, 1997), along with a 5th degree closure term and constant intensity offset. In order to decrease possible instrumental instabilities and to minimize the influence of stratospheric trace gases in the retrieval, the zenith spectrum from each scan was 40 used as a reference. Moreover, only data gathered in off-axis mode with SZA < 75° were used in this work.





Similarly, the $O_4$ differential slant column densities (dSCDs) were also retrieved (337-370 nm spectral window) in order to invert the vertical profile of the aerosols extinction coefficient (AEC) and therefore to characterize the scattering properties of the atmosphere and the light path of the photons reaching the detector. The reliability of the aerosol vertical information retrieved by MAX-DOAS observations has already been demonstrated under different visibility conditions (e.g., Frieß et al.,

2016). This retrieval is based on the concept that the concentration $O_4$ is known and stable in the atmosphere. Hence, a variation in the $O_4$ dSCDs is usually related to a change of the optical path, generally due to the presence of aerosols (e.g., Hönninger et al., 2004; Wagner et al., 2004). The inverted AEC vertical profile was then used as input for the linear inversion of the vertical profile of BrO vmr.

In this work, the inversion of the vertical profiles of the AEC and the BrO concentration was based on the Optimal Estimation

Method (e.g., Rodgers, 2000). The radiative transfer model (RTM) used was LIDORT (Spurr 2008) and the inversion scheme was BePRO (BIRA, Clémer et al., 2010). The procedure consisted on a two-step approach. First, the AEC were retrieved from the observed $O_4$ dSCDs through an iterative nonlinear process (e.g., Hendrick et al., 2014; Córdoba-Jabonero et al., 2016). The inferred AEC was then used to invert the targeted BrO profiles.

The RTM input parameters characterising the profile retrievals were carefully chosen for polar conditions and always bearing

in mind that the aim is to gain long-term observations in the hostile conditions of Antarctica. For the measurements performed from Marambio station, the pressure (P), temperature (T), $O_3$ and $NO_2$ vertical profiles were obtained from standard atmosphere for sub-arctic latitudes (Anderson, 1986). For the observations made from Belgrano, the considered P, T and $O_3$ profiles were obtained from monthly averaged available ozone-sonde records (from 1999 to 2006, e.g., Parrondo et al., 2014), while the $NO_2$ profile was taken from the same standard atmosphere. The modelled atmosphere was stratified into layers of

100 m from 0 to 4 km altitude, layers of 1 km from 4 to 6 km (wider grid related to less sensibility at those altitudes) and layers of the same width of those of the standard atmosphere above this altitude. The retrieved profiles were obtained up to an altitude of 6 km. In the inversion scheme, the diagonal elements of the measurement uncertainty covariance matrix were the squared of the dSCDs error after the DOAS fit ($1\sigma$). The diagonal elements of the a priori covariance matrix was calculated based on Franco et al. (2015), with a ß set as 0.8. The non-diagonal elements were calculated following a Gaussian distribution with a

correlation length of 100 m for aerosols and 300 m for trace gases, respectively (e.g., Hendrick et al., 2004). Therefore, the error of the retrieved profiles provided in this work contain the measurement error and the smoothing error of the retrieval.

Given that the measurements in Antarctic stations are frequently affected by blowing snow, the aerosol optical properties were obtained using Henyey-Greenstein phase functions for single scattering albedo SSA = 0.999982 and asymmetry parameter g = 0.89, corresponding to typical values of clean ice crystal (e.g., Frieß et al., 2011). After several tests considering typical

snow albedos (between 0.8 – 0.9), the surface albedo was set to 0.8. In order to avoid unrealistic values, an upper limit for aerosols optical depth (AOD) was set to 5, neglecting therefore all the observations made in such complicated conditions from the light scattering point of view. Also, only retrievals with the degrees of freedom higher than 1 were taken under consideration.

After the AEC vertical profile was estimated at 360 nm by means of the measured $O_4$ dSCDs, the aimed BrO vmr vertical

profiles were retrieved at 338 nm using the calculated AEC as input of the RTM. In order to properly include the inferred AEC in the retrieval of the BrO vmr profiles, they were calculated at the corresponding wavelength using an Angstrom parameter of 2.2 (e.g., Hegg et al., 2010). An exponential decreasing profile corresponding to AOD = 0.02 was chosen as a priori for the AEC vertical profile. The scale height (SH) of the a priori AEC was set to 0.5 km for Belgrano and to 2 km for Marambio, since these values provided the lower differences between observed and modelled $O_4$ dSCD for each station. The a priori BrO

vertical density corresponded to an exponentially decreasing profile with SH = 1 km and a surface value of ~1.5 pmol mol$^{-1}$.



### 2.3 Ancillary Data

In order to interpret the aimed bromine results, additionally to the spectra gathered by the MAX-DOAS measurements, near surface $O_3$ vmr measured at both stations were also compiled. At both sites, the surface $O_3$ was measured with ozone analyzers (Thermo Environmental Instrument, Thermo Fisher Scientific, model 49; i.e., TEI49). The operation principle of this in situ

instrumentation consists in the attenuation of an ultraviolet light beam (254 nm) by an air sample containing ozone and has a manufactured sensitivity and limit of detection of 1 nmol mol$^{-1}$. In the case of Marambio station, which contributes to the GAW network (Global Atmosphere Watch, WMO- World Data Centre for Greenhouse Gases WDCGG), the measurements of surface $O_3$ were carried out by the National Meteorological Service of Argentina (SMN) and can be retrieved from the WMO - World Data Centre for Greenhouse Gases (https://ds.data.jma.go.jp/gmd/wdcgg/). These data from the TEI49 from

Marambio is compared every year against the regional standard (WMO, RCC-BsAs, TEI49PS). At the research site of Belgrano, the year-round surface $O_3$ is measured by INTA since February 2007 (e.g., Jones et al., 2013). At this site, the inlet of the analyzer, protected from rain, snow and dust, is placed 0.85 m above the roof of the base in the cleanest area of the station, being free of pollution from the research site.

Additionally, the weather information was obtained from the observations performed by the SMN at Marambio (WMO station

89055) and by INTA at Belgrano. In the case of Marambio, the weather station is installed in the so-called Scientific Pavilion of the Marambio Antarctic Station together with an Automated Met Station (AMS), which measures the temperature, humidity, precipitation, wind speed and direction besides the atmospheric pressure. The data acquisition system is carried out through a Datalogger Cambell Scientific CR100. In the case of Belgrano, the weather parameters are gathered by a Vaisala weather station installed in the site in 2009. In this case, the weather station is installed on the roof of the base, in a 210 cm mast and it

provides wind speed and direction, atmospheric pressure, temperature and relative humidity.

### 3 Results and Discussions

This section is divided into three main parts. First, it presents the time series of the DOAS measurements and the ancillary observations performed during 2015, offering an overview of the information gathered within the frame of this study. Then, the details and discussion of the retrieved BrO and AEC vertical distributions at the two sites are provided. Finally, the

activation of bromine in the Antarctic troposphere during 2015 is investigated along with the reactivity of the Antarctic troposphere with regard to inorganic reactive bromine.

### 3.1 Time series

This section presents the observations gathered during 2015 at each station. It first shows the DOAS measurements in terms of aerosol optical depth (AOD) and BrO vertical column densities (VCD). Later on, it shows the results of the ancillary

observations (weather parameters and surface ozone).

### 3.1.1 DOAS observations: BrO VCD$_{2km}$ and AOD$_{2km}$

Herein we present the AOD and the BrO VCD measured with the DOAS technique during the sunlit period of 2015. This period lasted for about 8.5 months in Marambio and 7.5 months in Belgrano (Fig. 3 y 4). Due to instrumental issues, there were missing data at the beginning of the year at both stations. Note, however, that the polar sunrise was well covered at both

sites and therefore the peak season of the bromine activation (e.g., Simpson et al., 2007).

As mentioned in Sect. 2.2.1, the sensitivity of the MAX-DOAS observations decreases with altitude. Hence, aiming also at comparing both stations, here we refer to the AOD and BrO VCD inferred in the first 2 km of the troposphere at each site (i.e., AOD$_{2km}$ and VCD$_{2km}$, respectively). Figure 3 shows the AOD$_{2km}$ retrieved at both stations. Observations indicate that the aerosol optical thickness of the low troposphere at Belgrano was generally higher than at Marambio. In addition to this



geographical dependence of the AOD, Fig. 3 also suggests that the period with higher aerosol thickness lasted longer in the southernmost station of Belgrano. As seen in the figure, while high AOD were observed at Belgrano during most of the sunlit period, the AOD at Marambio intensified from September until December. The geographical variability of the aerosol load within Antarctica has already been reported (e.g., Savoie et al., 1993; Minikin et al., 1998), although further studies on the

distribution of aerosols are needed in order to understand the interannual variability of the different sources (e.g., Giordano et al., 2017). Further insights of the aerosol properties within the Antarctic troposphere during 2015 will be provided in a following work (Gómez-Martín et al., 2018, in preparation).

Regarding the aimed BrO, the pseudo-vertical column densities ($VCD_{2km}$) are calculated by integrating the BrO concentration obtained within the first 2 km of the troposphere. Results, shown in Fig. 4, indicate that BrO was present in the sunlit Antarctic

troposphere at both stations. The median BrO $VCD_{2km}$ values at both sites were quite similar (~ $0.5 \cdot 10^{13}$ molec $cm^{-2}$) and 75 % of the observations at both stations fell below a similar range ($0.8 \cdot 10^{13}$ molec $cm^{-2}$). Also, at both sites, the maximum BrO $VCD_{2km}$ values were observed after the polar sunrise and the BrO levels were undetectable just before the polar sunset and immediately after the polar sunrise. However, the magnitude and variability of the BrO maximums direct the difference between both stations, with maximum BrO $VCD_{2km}$ observed in Marambio being 3.2 times higher than in Belgrano. As can

be observed in Fig. 4, it is also worth noticing the clear photolytic activation of BrO in Marambio during austral spring, with levels an order of magnitude higher than the median BrO values at the station. Insights on the vertical distribution of BrO and aerosol extinction in the Antarctic troposphere are provided later on in Sect. 3.2.

### 3.1.2 Ancillary observations: meteorological parameters and surface $O_3$

Aiming at contextualizing both research stations, this section briefly presents some weather parameters and the near surface
ozone characterising the sites of Marambio and Belgrano during 2015.

Regarding the weather information at each station, the mean observed values of meteorological parameters such temperature (T) and precipitation are provided in Table 3. Overall, observations indicate that Belgrano station sits at a dryer and colder location where, in 2015, temperatures dropped below -40º C. Regarding the wind measurements, the speed observed at both stations is shown in Fig. 5, while the wind roses are given in Fig. 6. Observations indicate that, although the higher gusts of

wind were quite similar at both stations (~120 km $h^{-1}$), the mean wind speed at Marambio was in general 50% higher than in Belgrano. Concerning the wind direction observed at each station during 2015 (Fig. 6), while the air masses arriving at Marambio had no clear dominant direction, those arriving at Belgrano came usually from the south-south west. Hence, based on the wind rose (Fig. 6), during 2015 Marambio was mostly influenced by air masses coming along the west edge of the Weddell Sea but also from the surrounding Scotia and Amundsen Seas, while the air masses from continental Antarctica

dominated the observations performed at Belgrano station.

As for the near surface $O_3$, the annual variation at both stations showed the seasonal pattern expected in high-latitude stations (Antarctic and sub-Antarctic regions, Fig. 7) and is typical of remote, low NOx environments with $O_3$ being accumulated during winter (maximum) and destroyed (minimum) during summer. Also, the observed amplitude of the surface $O_3$ annual cycle at both stations were also characteristic of an Antarctic station (e.g., Helmig et al, 2007, Legrand et al., 2016). While in

2015 the median values of surface $O_3$ were quite similar at both locations (23 nmol $mol^{-1}$ at Marambio and 24 nmol $mol^{-1}$ at Belgrano), the maximum values reported at Marambio (36.8 nmol $mol^{-1}$) were about a 10 % higher than those observed at Belgrano station. Regarding the absolute minimum surface $O_3$ detected, 2015 observations at Marambio indicate ozone depletion events (ODEs) with measurements very close or below instrumental detection limit (1 nmol $mol^{-1}$), while the minimum surface $O_3$ detected in Belgrano was not lower than 6 nmol $mol^{-1}$. This suggests that, compared to Belgrano,

Marambio is either as a more photochemically active region or a region more exposed to ozone depleted air masses. Noteworthy is also the high variability of surface ozone observed at Marambio during the polar sunrise as compared to





observations at Belgrano station. This behaviour is characteristic of a coastal Antarctic station as reported by e.g. Helmig et al. (2007).

## 3.2 Vertical profiles of BrO in the Antarctic troposphere

The time series of BrO vmr measured during 2015 in the first 6 kilometres of the troposphere of Marambio and Belgrano are
provided in Fig. 8, along with the AEC. One characteristic of these BrO observations is that, unlike conclusions of previous Antarctic studies suggesting the presence of reactive bromine above 4 km altitude (e.g., Frieß et al., 2004; Roscoe et al., 2014), during 2015 no elevated plumes of BrO were observed at either of the two Antarctic stations referred to in this work. In fact, during 2015, most of the BrO was confined within the first 2 km of the troposphere (Fig. 8). For clarity, Fig. 9 shows in more detail some examples of the time evolution of BrO as measured at Belgrano and at Marambio. The wind speed on those selected
days was below 10 m s$^{-1}$. As can be seen in Fig. 9, the maximum of BrO was located close to the surface although its specific altitude depended on the day, located always bellow 1 km of altitude. Some days the peak of BrO was located just above the surface (e.g., 11$^{th}$ November in Belgrano or 25$^{th}$ September in Marambio in Fig. 9), while in others that BrO maximum was slightly elevated suggesting heterogeneous reactions aloft (e.g., 29$^{th}$ October in Belgrano or 28$^{th}$ November in Marambio in Fig. 9). Worth noticing is also the time and seasonal variability of the occurrence of the maximum of BrO vmr. As shown in
Fig. 9, on e.g. 29$^{th}$ of October in Belgrano, some days BrO vmr followed the diurnal evolution with a noon maximum predicted by model studies (e.g., Saiz-Lopez et al., 2008) and observations (e.g., Buys et al., 2013) where the BrO formation is linked to the solar irradiance and the photolysis of bromine sources (e.g., Br$_2$, BrCl). On the other hand, e.g. on 28$^{th}$ of November in Marambio (Fig. 9), on other days BrO was present in the low troposphere with a double maximum (morning and evening), characteristic of a late spring behaviour and more related to bromine being recycled e.g. through HOBr (e.g., von Glasow et
al., 2002; Pöhler et al., 2010; Liao et al., 2012; Buys et al., 2013). This time and altitude dependence of the BrO distribution in the troposphere reinforces the benefits of the sort of instrumentation employed in this work, which offers vertically resolved information and is able to perform long-term observations.

Figure 10 shows a summary of the BrO vmr vertical profiles observed at each station during the sunlit period of 2015. The median BrO vmr in the lowest layers of the troposphere (< 0.5 km) was similar at both stations (~1.6 pmol mol$^{-1}$ above the
surface) with 75 % of the BrO data below 2.5 pmol mol$^{-1}$. However, as shown in Fig. 8, the maximum BrO values observed after the polar sunrise at Marambio (26.0 ± 0.4 pmol mol$^{-1}$) were over a threefold of those observed after the sun rose at Belgrano (see also Fig. 4). This maximum BrO was detected during austral spring at ~ 200 m of altitude, with a magnitude dependent on the station. This slightly elevated peak of BrO (e.g., Fig. 10) and mentioned also above, has already been foreseen by studies accounting for the vertical gradient of the acidity of the aerosols and/or the effect of convection (e.g., von Glasow
et al., 2002; Wagner et al., 2007). Also note that, as shown in Fig. 8, while in Belgrano the maximum BrO observed during October-November (8.1 ± 0.6 pmol mol$^{-1}$) quadrupled its mean value measured during the rest of the sunlit period, the BrO values observed in Marambio just after the sunrise were over 15 times higher than the BrO mean values at that station. All this suggests that the halogen reactivity at Marambio is considerable stronger than at Belgrano (see also Sect. 3.3). The BrO vmr ranges reported herein (Fig. 10) are comparable to previous tropospheric Artic studies (e.g., Tuckermann et al., 1997;
Hönninger and Platt, 2002; Prados-Roman et al., 2011; Liao et al., 2012; Peterson et al., 2017; Simpson et al., 2017) and consistent with the few existing Antarctic measurements (e.g., Table 1). By adding the BrO measurements provided in the frame of this work to the few previous ground-based observations performed at Antarctica from different sites, Fig. 11 depicts an updated map of the maximum values of BrO observed in the lower troposphere of Antarctica, pointing once more its heterogeneity with regard to reactive bromine load. Section 3.3 offers a closer look to this heterogeneity.
Overall, the observations presented in this study indicate that the vertical profile of BrO in the Antarctic troposphere descended with altitude, with no noticeable detection above 2 km (Fig. 10). Note that, in this work, the detection limit of BrO is regarded as the threshold value above which the inferred BrO is significantly higher than the noise of the inversion. In this case, it is





defined as double of the inversion error (Sect. 2.2.2) and corresponds to a mean value of 1 pmol mol$^{-1}$. Note that, although the definition of the height of the boundary layer over ice and snow surfaces (e.g., Anderson and Neff, 2008) is out of the scope of this work, the BrO detection limit here provided may be regarded as an upper limit of BrO in the free troposphere since former studies place the top of the boundary layer in Antarctica between 100 m and 2 km, depending on the boundary layer

parametrization and time of the year (e.g., King et al. , 2006; Nygård et al., 2013). This upper limit of BrO in the free troposphere of Antarctica is consistent with the few previous studies of the vertical distribution of this trace gas in the Arctic and Antarctic regions that set upper limits of BrO in the polar free troposphere of 1.5 and 2 pmol mol$^{-1}$, respectively (e.g., Frieß et al., 2011; Prados-Roman et al., 2011; Peterson et al., 2017; Hüneke et al., 2017).

In addition to the BrO knowledge gained after this work, noteworthy is also the information related to the vertical AEC (Fig.

8, lower panels), sustaining the particularity of the surroundings at each station. In addition to the aforementioned different aerosol optical thickness at both stations (Sect. 3.1.1), there is also a noticeable difference regarding the seasonality and altitude of the maximum AEC at the two sites. While at Marambio the peak of the AEC appeared close to the surface with a clear maximum extinction observed in November, the observations performed from Belgrano suggest that, at this site, the height of the aerosol layer was much more variable than at Marambio, manifesting once more the relevance of vertically resolved

observations within the Antarctic troposphere. As mentioned before, the work of Gómez-Martín et al. 2018 (in preparation) will address these issues.

### 3.3 BrO$_x$ in Antarctica

In order to investigate the hinted heterogeneity of the Antarctic lower troposphere regarding reactive bromine (BrO$_x$ = Br + BrO), in this section the budget of bromine [Br] is estimated considering steady state of BrO in a pristine atmosphere with

virtually no NO (e.g., Zeng at al., 2006) and a concentration of ClO, HO$_2$ and OH of 1.7·10$^8$ molec cm$^{-3}$, 2.2·10$^7$ molec cm$^{-3}$ and 3.9·10$^5$ molec cm$^{-3}$, respectively (e.g., Halfacre et al., 2014; Bloss et al., 2007). Hence, [Br] can be estimated from the observed BrO and O$_3$ concentrations as (e.g., Hausmann and Platt, 1994; Le Bras and Platt, 1995; Zeng et al., 2006; Stephens et al., 2012):

$$[Br] = [BrO] \frac{2\, k_{BrO+BrO}\, [BrO] + k_{BrO+ClO}\, [ClO] + k_{BrO+HO_2}[HO_2] + k_{BrO+OH}\, [OH] + J_{BrO}}{k_{Br+O_3}\, [O_3]} \quad (2)$$

where $J$ represents the rate of photolysis and $k$ the different reaction rates (Table 4). Since the measurements of O$_3$ were performed near the surface, accordingly only the BrO retrieved in the lowest atmospheric grid (i.e., 100 m) is considered for the calculation of the Br and BrO$_x$ budgets at each site. The possible influence of horizontal advection and blowing snow is limited in the data set by applying an upper limit for the wind speed of 6 m s$^{-1}$. Note that previous studies pointed 8 m s$^{-1}$ as the wind threshold for blowing snow (e.g., Jones et al., 2009) and others indicated that the steady state approximation is valid

for wind speeds lower than the 6 m s$^{-1}$ threshold considered in here (e.g. Liao et al., 2012).

Figure 12 shows the 2015 seasonal evolution of the BrO$_x$ budget at each station and Fig. 13 presents the BrO-Br-BrO$_x$ statistical analysis in the form of box charts at each site. Since the kinetic calculations used herein are based on observations performed under low wind conditions, these budgets may be considered as representative of the surroundings of each station. Figure 12 indicates that, in agreement with Peterson et al. (2015), the presence of reactive bromine at both stations do not only correspond

to advected bromine-enriched air masses or blowing snow. As expected from previous polar studies (e.g., Simpson et al., 2007 and references therein) and shown in Fig. 12, the maximum bromine-related reactivity of the troposphere at both stations takes place just after the photolysis is triggered with the polar sunrise. As shown in Fig. 12, this maximum reactivity does occur at a medium O$_3$ regime at both stations (10-25 nmol mol$^{-1}$). The study of the BrO-Br-BrO$_x$ data (Fig. 13) indicates that, during the sunlit period of 2015, the mean budget of BrO$_x$ at Belgrano (2.0 pmol mol$^{-1}$) was ~ 17 % higher than at Marambio. However,

just after sunrise, the BrO$_x$ budget (and hence the troposphere reactivity) at Marambio triplicated the one at Belgrano (e.g., Fig. 12). Estimated values for atomic bromine radical present in the lowermost troposphere during the sunlit period of 2015



were up to 1.4 pmol mol$^{-1}$ at Belgrano and up to 3.4 pmol mol$^{-1}$ at Marambio (Fig. 13). These ranges are in line with previous model studies for Antarctic latitudes (e.g., von Glasow et al., 2004; Saiz-Lopez et al., 2008) and in the lower limit of Artic model studies (e.g., Thompson et al., 2017).

Overall, these estimations indicate that the BrO$_x$ partitioning was clearly driven by BrO at both sites, indicating that ozone in
general was not fully depleted as confirmed by the observations (Sect. 3.1.2). The evolution of the ratio Br to BrO after the polar sunrise is shown in Fig. 14 for each site. The initial Br : BrO after dawn was ~ 0.05 at both stations. Throughout the polar spring, during ODEs, that ratio raised over fourfold at both sites. The baseline of the ratio Br to BrO ratio during the sunlit period could be approximated by an exponential growth with a time constant of about 10 days in Belgrano and 17 days in Marambio (blue line in Fig. 14). Towards summer, that baseline increased up to 0.17 at Belgrano and to 0.10 at Marambio. In
the simplified scheme suggested by Eq. (2) and discussed in this section, this Br : BrO increase could be explained by the overall summer decreased of surface O$_3$ compared to springtime (Fig. 7). Additional investigations on the variability and geographical distribution of the bromine source gases throughout the year are suggested to address the bromine pathways in the Antarctic troposphere and their consequences. Bearing in mind this simplified schemed, based on Eq. (1), the bromine-mediated ozone loss rate can be assessed at each research site for the different BrO$_x$ regimes observed at low wind speed.
Similar median BrO values measured during 2015 at both stations (1.6 pmol mol$^{-1}$) yield similar ozone loss rate of 0.4 nmol mol$^{-1}$ day$^{-1}$ at both sites. During the bromine more active season of October-November at Belgrano (e.g., Fig. 12), this rate speeds up to 2.9 nmol mol$^{-1}$ day$^{-1}$. During September at Marambio (peak bromine season at that station), the bromine-mediated ozone loss occurs at a much faster rate between 0.7 and 17.4 nmol mol$^{-1}$ day$^{-1}$ (i.e., up to 6 times faster than at Belgrano). Former works have estimated that the bromine driven ozone loss in the polar atmosphere represents 44% of the total O$_3$
chemical loss (e.g., Liao et al., 2012; Thompson et al., 2017). Therefore, in the sites referred to in this work the shortest (i.e., at highest BrO$_x$ and low wind speed) ozone chemical lifetime $\tau_{O3}$ expected is 2.6 days at Belgrano and 0.7 days at Marambio. Further studies including different sources and sinks of bromine radicals in the Antarctic environment would be needed to confirm these numbers which herein are based on ozone depletion through (only) the BrO-BrO and BrO-ClO channels, dominant however in the polar spring (e.g. Simpson et al., 2007).
All these kinetics approximations are historically based on conclusions after numerical models and laboratory and campaigned-based observations obtained in the polar regions (mainly the Arctic; e.g., Simpson et al., 2007). Nevertheless, the year-round erratic behaviour of the wind speed in Antarctica at each station makes complicated the verification of these estimated (low wind) $\tau_{O3}$ with observations. However, the exemplary days provided in Fig. 9 with higher BrO at each station (upper figures) may serve the purpose (low wind speeds). For instance, based on the ozone observations (Fig. 7), the rate of O$_3$ depletion
measured at Marambio (25$^{th}$ September) was 4.1 nmol mol$^{-1}$ h$^{-1}$ and at Belgrano (29$^{th}$ October) it was of 0.58 nmol mol$^{-1}$ h$^{-1}$. Therefore, as suggested by the above related theoretical calculations, the destruction of surface O$_3$ during the bromine peak season was indeed much faster at Marambio (7 times faster than in Belgrano). Considering the mean O$_3$ vmr observed at each station on those days, the observed $\tau_{O3}$ at Belgrano was of 1.3 days while at Marambio it was tenfold shorter. Note that, as shown in Fig. 9 (upper figures), on those specific days the BrO load at Marambio was also over an order of magnitude higher
than at Belgrano. Comparing these observed $\tau_{O3}$ with the $\tau_{O3}$ estimated above from kinetics, the measurements show shorter $\tau_{O3}$ at both stations (50 % shorter at Belgrano and 18 % shorter at Marambio). The resemblance of the observed and calculated $\tau_{O3}$ at Marambio suggests that the assumptions made at Marambio's surrounding (e.g., the Br-Cl channel dominates the ozone depletion) is close to reality which seems not to be the case for Belgrano's surroundings. This reinforces, once more, the need of further investigations for a better understanding of all the processes and key parameters involved in the halogens' pathways
in the Antarctic troposphere.





## 4 Summary and Outlook

As a result of its remoteness and a more complex logistics compared to the Arctic region, the characterization of the Antarctic troposphere with regard to halogen compounds is still very scarce. This study reports on the presence and distribution of reactive inorganic bromine ($BrO_x$) in the Antarctic troposphere. Prior to this study, throughout Antarctica only five sites had

reported ground-based observations of BrO in the low troposphere. With the appropriate instrumental set up in the research stations of Belgrano and Marambio, INTA has expanded this net considerably. Moreover, to the authors' knowledge, this is the first study where these bromine observations are reported simultaneously from two Antarctic stations making possible to gain an insight into the geographical distribution of reactive bromine in the Antarctic troposphere. Additionally, through the 2015 MAX-DOAS measurements performed from the two sites, this work presents vertically resolved observations of BrO at

two different Antarctic stations with a dedicated inversion scheme for inferring the vertical distribution of BrO throughout the Antarctic troposphere. Furthermore, the aerosol extinction and the surface ozone at the two sites are also provided.

Overall, results show the expected seasonal and daytime variation of BrO related to the photolytic activation of reactive bromine triggered by the polar sunrise at the two sites. However, as referred above and unlike some former studies, during the sunlit period of 2015 no elevated plumes of BrO were detected above 2 km. In fact, in line with previous polar studies, this

work sets an upper limit of BrO in the free troposphere of Antarctica of 1 pmol mol$^{-1}$. Also, this study reports on the positive detection of BrO in the low troposphere (< 2 km) of Antarctica even under low wind conditions, suggesting that the presence of this trace gas is not only related to horizontal advection and pointing towards surface emissions and vertical mixing. As for the vertical and geographical distribution of BrO in the lower layers of the troposphere, observations indicate a slightly elevated BrO peak at 200 m at both stations, with a maximum value measured at Marambio considerably higher than the observed at

Belgrano (26 pmol mol$^{-1}$ vs. 8 pmol mol$^{-1}$, respectively).

In general, the observations and assessments presented in this work reveal a remarkable heterogeneity of the Antarctic low troposphere with regard the budget of reactive bromine. Beyond blowing snow, the inferred threefold enhancement of $BrO_x$ at Marambio compared to Belgrano after the polar sunrise denotes a geographical heterogeneity also on the bromine sources. Marambio sits on a region surrounded by open waters and seasonal sea ice while the dominant sea ice nearby Belgrano is

perennial. Since bromine explosions are linked to heterogeneous reactions related to e.g. sea ice, open leads and snow surfaces, the type of sea ice and its seasonal evolution around each station may be a good starting point to tackle the bromine sources riddle and to investigate how climate change may affect the budget of $BrO_x$ in the troposphere of Antarctica. Moreover, the geographical distribution of $BrO_x$ and its partitioning addressed in this work also suggests that the reactivity of the troposphere at Marambio is particularly enhanced compared to other Antarctic sites ("hot spot"). Since the presence of $BrO_x$ in the polar

atmosphere represents a sink for elemental mercury, this study also reveals the tip of the Antarctic Peninsula (Marambio) as a region for potentially enhanced mercury deposition (bioaccumulation) worth looking into. Also, dedicated investigations combining collocated observations of e.g. halogenated substances (not only BrO), organic compounds, DMS, $NO_x$, $HO_x$, particles and sea ice properties at different stations, could assist a thorough study of the bromine sources and pathways at Antarctica, their geographical distribution and their projections under a changing environment.

Besides the bromine-related information gained after work, this study also emphasizes the benefits of deploying quality instrumentation in pristine and remote locations able to provide not only surface but also vertically resolved information. It also shows the scientific benefits of maintaining long-term observations despite the efforts related to sustaining research activities in such hostile environment. The data provided by the two ground-based instruments presented herein may, for instance, assist the satellite retrievals to distinguish between tropospheric and stratospheric BrO signal and hence facilitate a

more accurate assessment of e.g. stratospheric BrO and ozone trends. Additionally, they could also serve chemistry-climate models for constraining the chemistry behind processes specifically related to polar regions, areas where global models are often weak (particularly in Antarctica).





*Data availability.* Data are available upon request from the corresponding author.

*Competing interests.* The authors declare that they have no conflict of interest.

**Acknowledgements**

Sea ice concentration maps are obtained from the website of the Sea Ice Remote Sensing group of the University of Bremen (https://seaice.uni-bremen.de/start/data-archive/). This work was supported by the Spanish Ministry of Economy and Competitiveness (MINECO) under the projects HELADO (CTM2013-41311-P), VIOLIN (CGL2010-20353) and the Spanish contribution to ORACLE-O3 (POL2006-00382). Authors would also like to acknowledge the work done by the different

technicians at the stations of Belgrano and Marambio.

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





**Table 1. Summary of the published ground-based observations of tropospheric BrO made at Antarctica.** Published
5 works of tropospheric BrO observations performed from different Antarctic stations. The time periods of the observations, measurements technique used and maximum BrO reported are also included. The "~" in the maximum BrO reported correspond to estimated values. For details, please refer to respective publication.

| Publication | Station | Period of the measurements reported | Measurement technique | BrO vmr (maximum pmol mol$^{-1}$) |
|---|---|---|---|---|
| Kreher et al. (1997) | Arrival Heights (77.8º S, 166.7º E) | Autumn and spring 1995 | Zenith sky DOAS | ~30 |
| Frieß et al. (2004) | Neumayer (70.6º S, 8.2º W) | 17 days during spring 1999 and 17 days during spring 2000 | Zenith sky DOAS | ~13 |
| Schofield et al. (2006) | Arrival Heights (77.8º S, 166.7º E) | 1 month and 22 days during spring 2002 | Zenith sky and Direct Sun DOAS | ~13 |
| Saiz-Lopez et al. (2007) | Halley (75.6º S, 26.5º W) | 4 seasons 2004/2005 | LP-DOAS | 20 |
| Buys et al. (2013) | Halley (75.6º S, 26.5º W) | 38 days during spring 2007 | CIMS | 13 |
| Grilli et al. (2013) | Dumont d'Urville (66.7º S, 140º E) | 4 days during summer 2011/2012 | CEAS | < 2 |
| Roscoe et al. (2014) | Halley (75.6º S, 26.5º W) | 2 months and 4 days during spring 2007 | MAX-DOAS | ~25 |
| Frey et al. (2015) | Dome-C (75.1º S, 123.3º E) | 1 month during summer 2011/2012 | MAX-DOAS | ~2-3 |
| This work | Marambio (64.2º S, 56.6º W) | 3 seasons 2015 (no winter) | MAX-DOAS | 26.0 |
| This work | Belgrano (77.9º S, 34.6º W) | 3 seasons 2015 (no winter) | MAX-DOAS | 8.1 |

**Table 2. Details of the MAXDOAS instruments installed in Antarctica.** The NEVAII instrument is located in Belgrano research station while the NEVAIII instrument is placed in Marambio.

| | Belgrano (NEVA II) | Marambio (NEVA III) |
|---|---|---|
| Spectrometer | TRIAX 180 | MicroHR |
| CCD | HAMAMATSU S7031-1008 | |
| Spectral resolution (nm) | 0.6 | 0.5 |
| Azimuth viewing angle (°) | 62 | 116 |
| Elevation angles (°) | 2, 3, 5, 10, 15, 30, 60, 90 | 1, 2, 3, 5, 10, 20, 30, 90 |




**Table 3. Meteorological parameters (temperature T and snow fall) observed during 2015 at both stations.** Data are provided by the World Meteorological Organisation (WMO), the Argentinian Meteorological Centre and INTA's meteorological station (Belgrano).

| Station | Mean T (ºC) | Maximum T (ºC) | Minimum T (ºC) | Coldest month | Mean T in coldest month (ºC) | Days with snow fall (%) |
|---|---|---|---|---|---|---|
| Marambio | -8.2 | 17.4 | -29.9 | September | -16.8 | 60.8 |
| Belgrano | -13.7 | 3.0 | -43.9 | August | -19.7 | 47.7 |

**Table 4. Rates (*k*) of reactions** provided in Sect. 1 and employed in Sect. 3.3. The temperature used for the calculations was T = 262 K, similar to the mean temperature observed during 2015 at each station (Table 3).

| Reaction | Rate constant ($cm^3$ $molec^{-1}$ $s^{-1}$) | Reference |
|---|---|---|
| $Br + O_3$ | $8.02 \cdot 10^{-13}$ | Sander et al. (2006) |
| $BrO + BrO$ | $3.54 \cdot 10^{-12}$ | Sander et al. (2006) |
| $BrO + ClO$ | $7.83 \cdot 10^{-12}$ | Atkinson et al. (2007) |
| $BrO + HO_2$ | $3.03 \cdot 10^{-11}$ | Atkinson et al. (2007) |
| $BrO + OH$ | $4.67 \cdot 10^{-11}$ | Atkinson et al. (2007) |





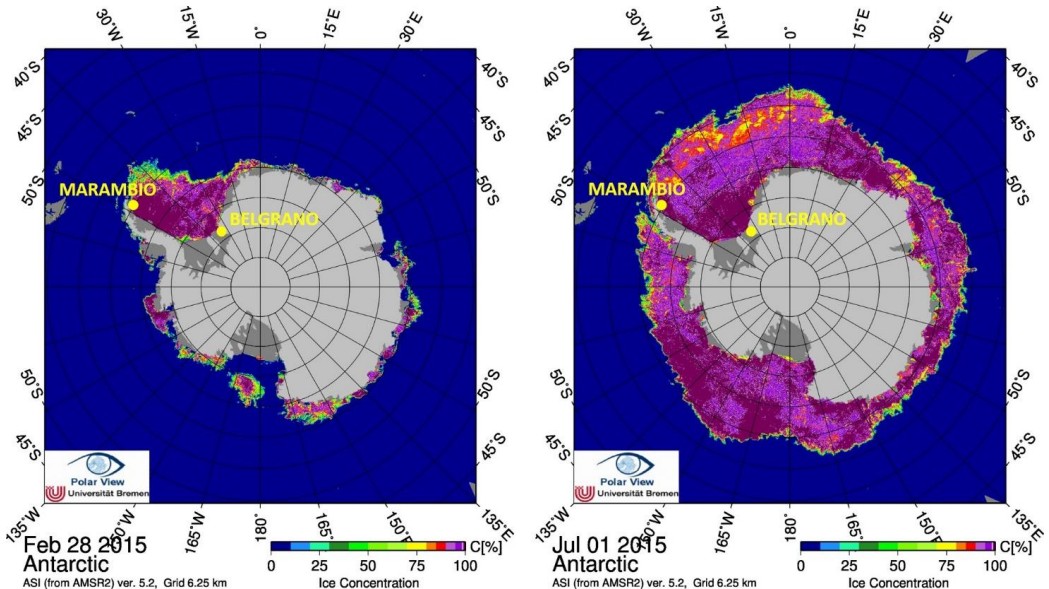

**Figure 1: Sea ice concentration surrounding the two Antarctic Stations.** The figure shows the sea ice concentration in Antarctica at the end of the austral summer (left) and at mid-winter (right) of 2015. The sea ice maps are downloaded from https://seaice.uni-bremen.de/databrowser/ (Spreen et al., 2008). The two Antarctic stations of Marambio and Belgrano are marked in yellow in both figures. Note the variability of the sea ice mainly nearby Marambio.

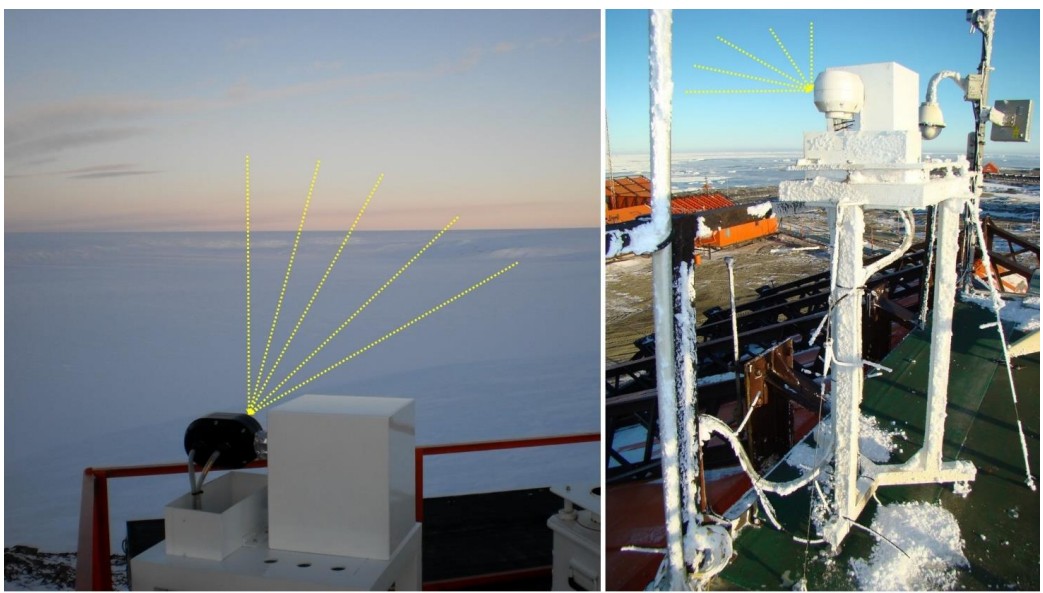

**Figure 2: INTA's MAXDOAS instruments mounted in the two Antarctic Stations.** The outdoor unit of the MAXDOAS instrument installed in Belgrano is shown in the left figure while the one in Marambio is shown in the right. By scanning the atmosphere at different elevation angles (yellow lines), vertical information of aerosols and trace gases can be retrieved.





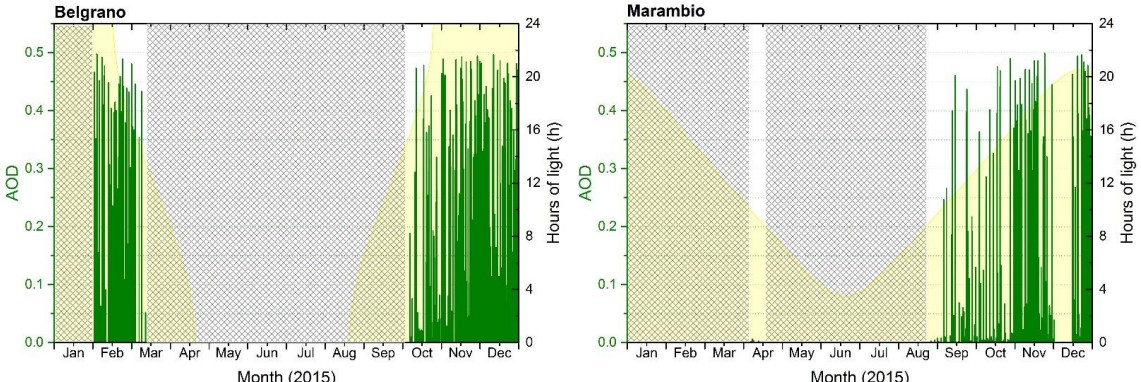

**Figure 3: Time series of the aerosol optical depth (AOD) in the first 2 km of the troposphere as observed in Belgrano (left) and in Marambio (right) during 2015.** The horizontal scale indicates the time of the year while the left vertical scale shows the AOD. The scale in the right shows the hours of light at each station (shown in yellow in the plots). Note the same scales apply to both figures. Time periods without MAXDOAS observations are indicated with shaded areas.

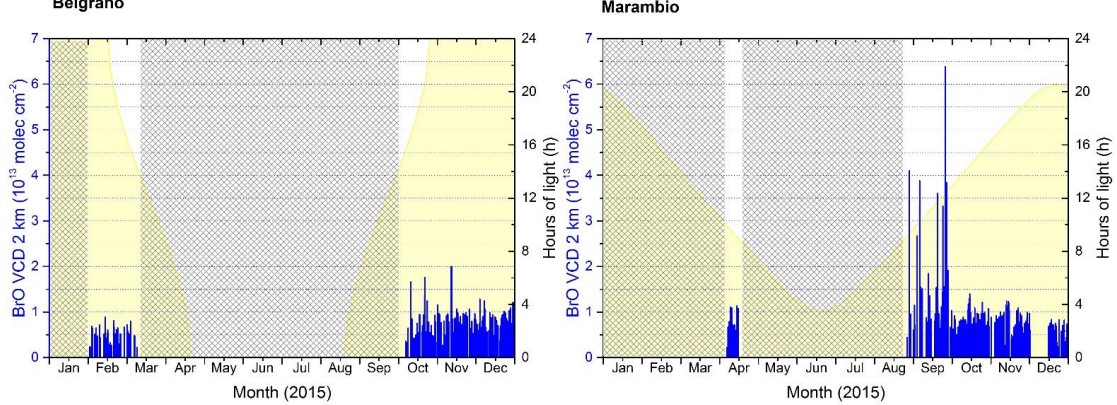

**Figure 4: Time series of the BrO vertical column density (VCD) in the first 2 km as observed in Belgrano (left) and in Marambio (right) during 2015.** The horizontal scale indicates the time of the year while the left vertical scale shows the BrO VCD. The scale in the right shows the hours of light at each station (shown in yellow in the plots). Note the same scales apply to both figures. Time periods without MAXDOAS observations are indicated with shaded areas.




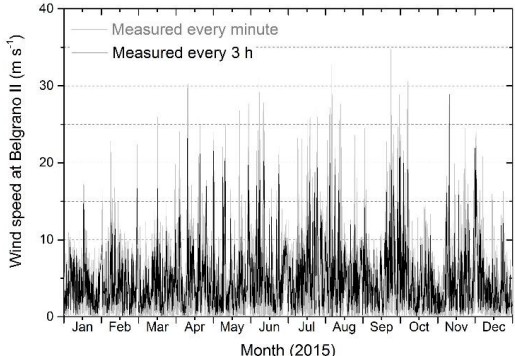
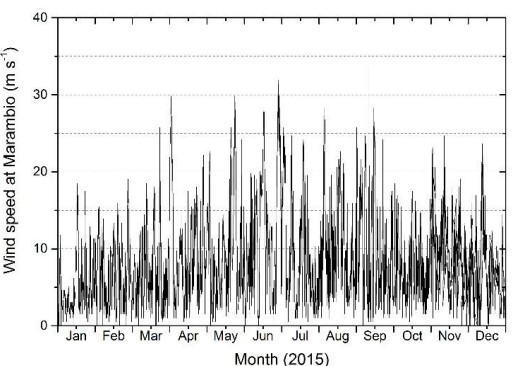

**Figure 5: Wind speed at Belgrano (left) and at Marambio (right) during 2015.** The data at Marambio (right figure) derive from the observations performed every 3 h by the Argentinian Weather Service (WMO Station 89055). The data at Belgrano (left figure) are gathered by INTA's weather station place at the site. The Belgrano's raw data (every minute) are shown in grey while the measurement gathered every 3 h in black which is Marambio's data frequency.

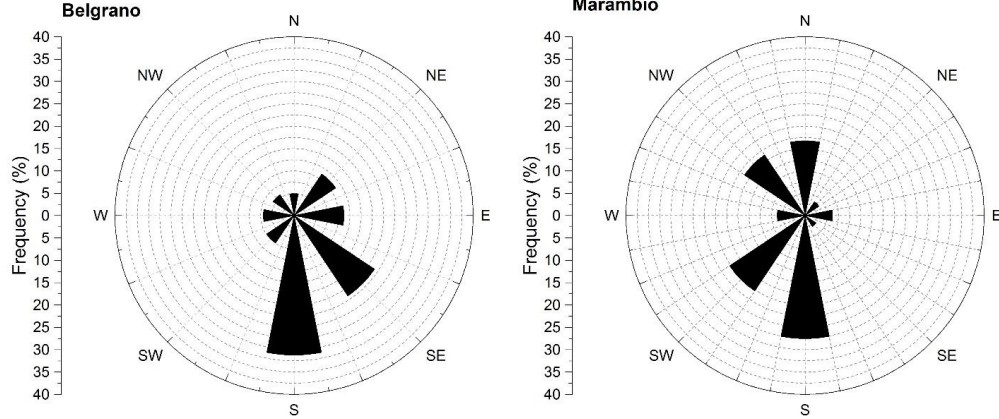

**Figure 6: Wind rose at Belgrano (left) and Marambio (right) stations (2015).** The vertical scale indicates the frequency count. The Belgrano's data are gathered form INTA's weather station and those at Marambio are provided by the WMO and the Argentinian Weather Service.





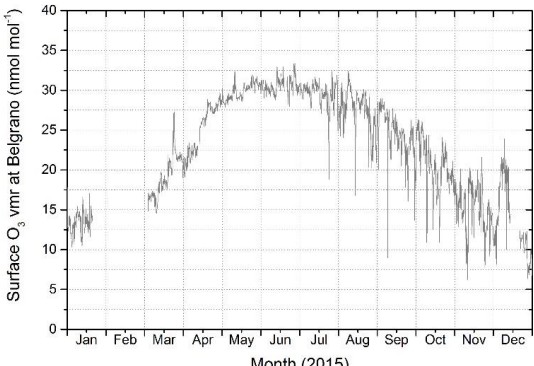 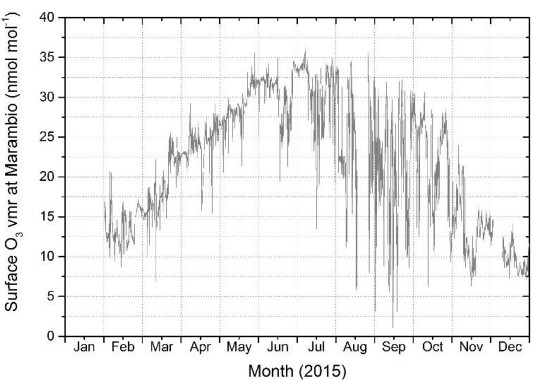

**Figure 7: 2015 near-surface ozone observations at Belgrano (left) and Marambio (right) research stations.** Note the same vertical scale
15   in both plots. Both data set were gathered by in situ O₃ observations made by INTA (at Belgrano) and by the Argentinian Meteorological
Service (at Marambio).




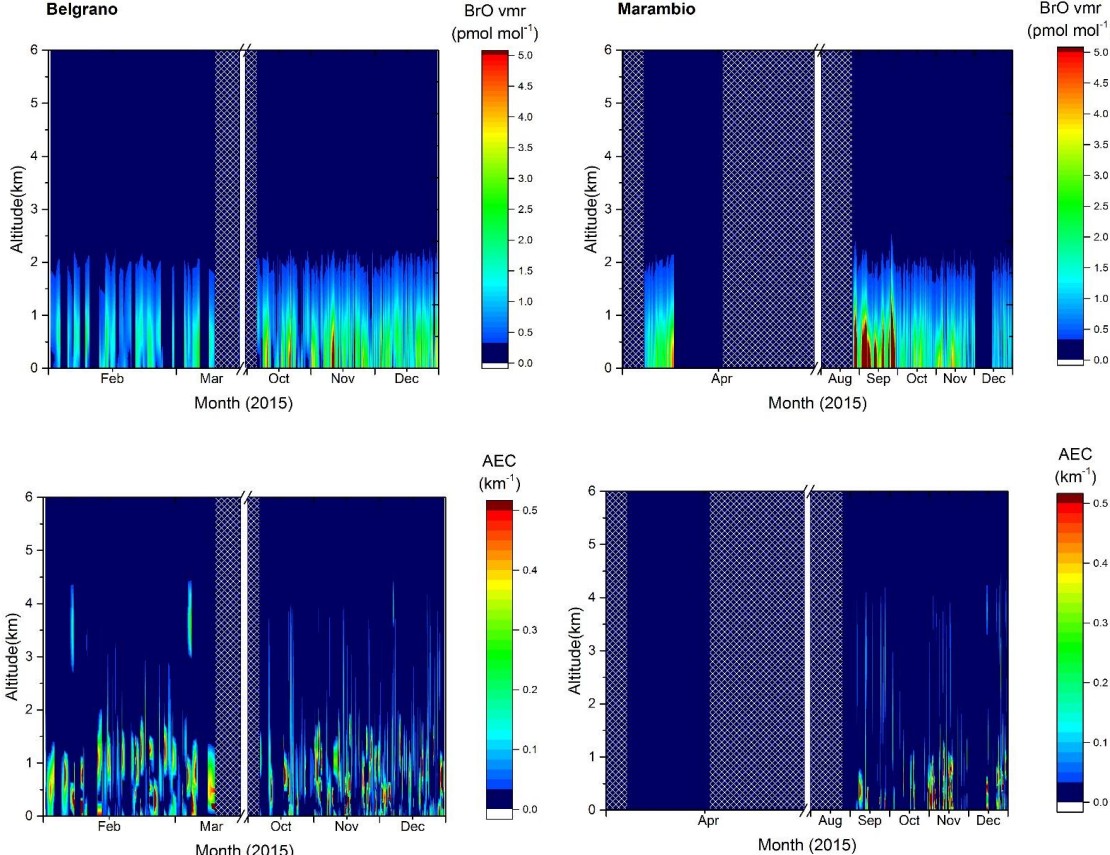

**Figure 8: BrO vmr and AEC observed during 2015 in the troposphere of Belgrano (left) and Marambio (right).** The vertical scales
show the altitude and the horizontal scales indicate the periods of the measurements, which depended on the station. The color code of the
upper figures corresponds to the BrO vmr and are forced to be the same for the sake of comparison. The same applies to the color code of
the lower figures that indicate the AEC at each station. The BrO vmr higher than 5 pmol mol$^{-1}$ are shown in dark red. Time periods with no
observations are indicated with the shaded areas.



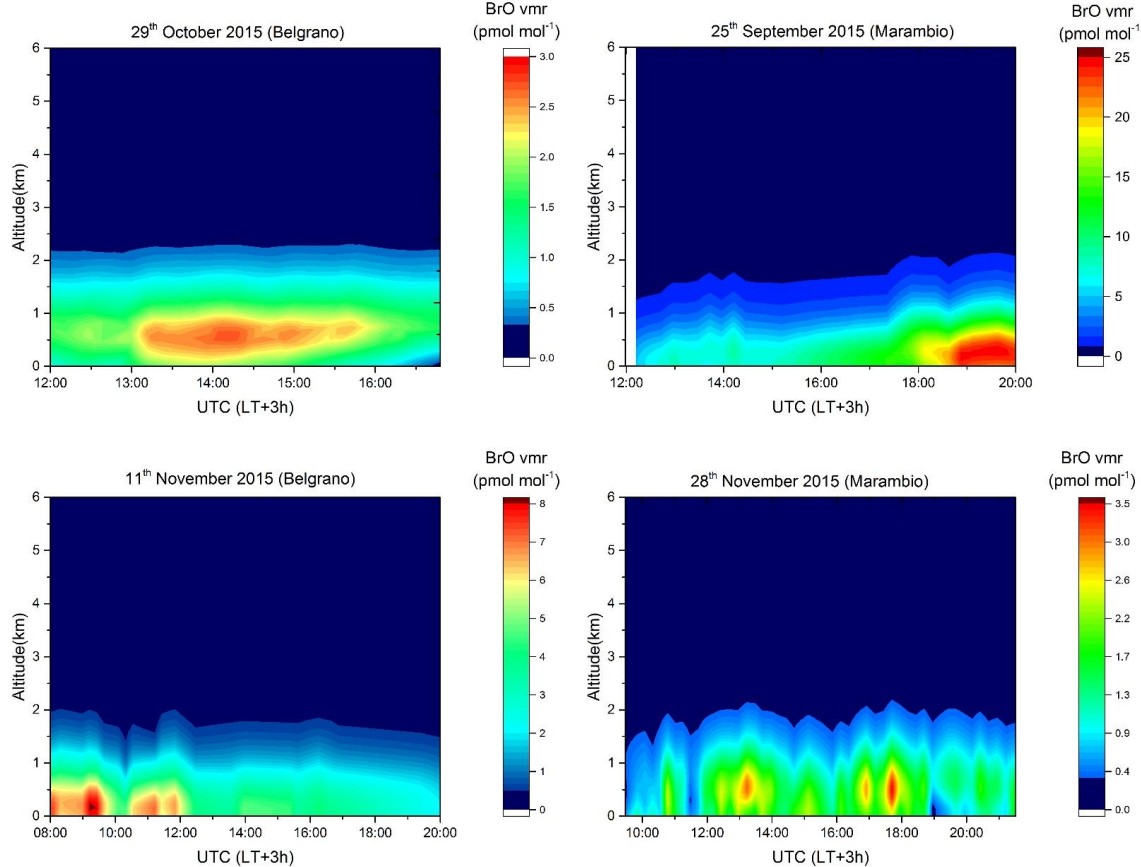

**Figure 9: Examples of the daily and vertical evolution of BrO at Belgrano (left) and at Marambio (right).** The horizontal scales indicate the time of the day while the vertical scales show the altitude. The color code corresponds to the BrO vmr. Note the different scales of each day.



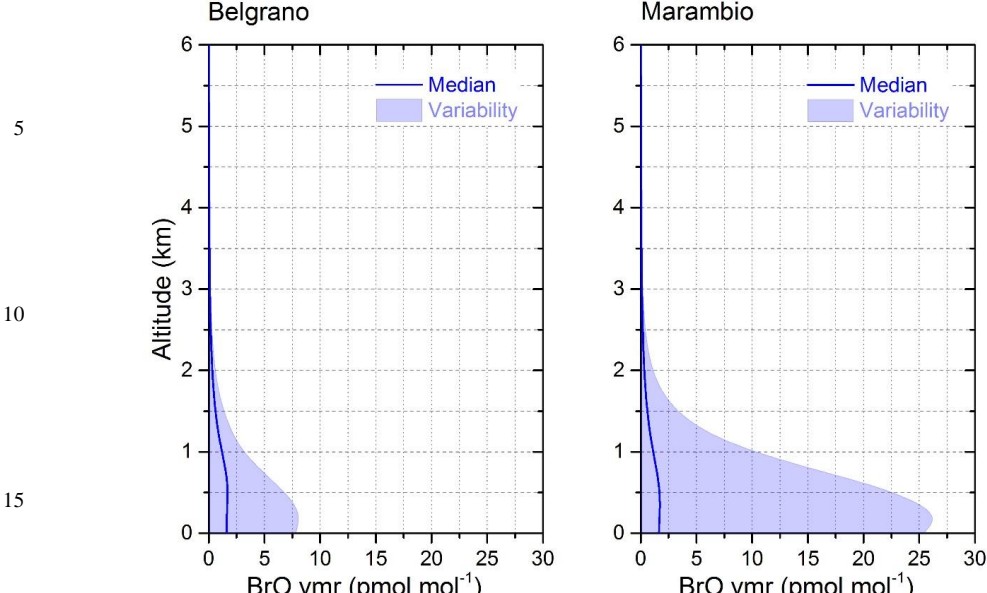

**Figure 10: Vertical profile of the BrO volume mixing ratio in the Antarctic troposphere (2015).** The observations performed from Belgrano station are shown in the left figure while those performed from Marambio are given in the right one. The median BrO values observed are indicated in thick blue lines while the shaded blue areas mark the variability range of BrO vmr throughout the sunlit period. Note the same scales in both plots.





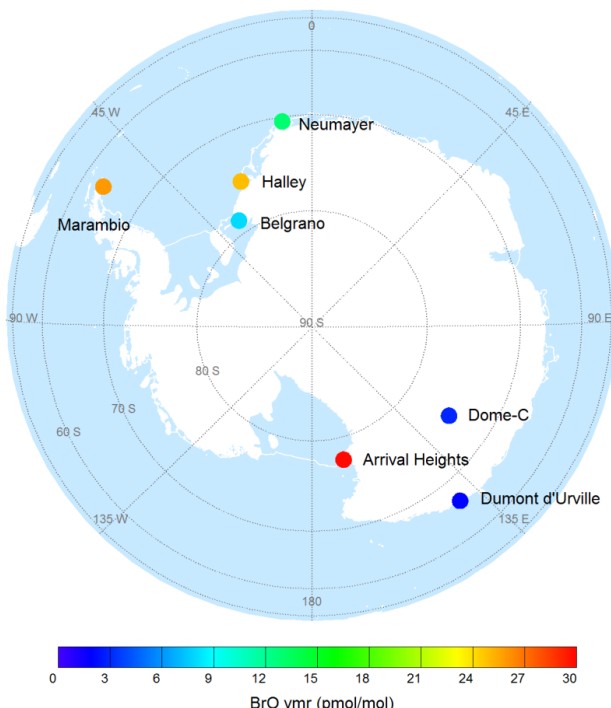

**Figure 11: Maximum values of BrO vmr reported in the low troposphere of Antarctica as measured by ground-based observations.**
The different sites where BrO has been reported in the low troposphere are indicated with a coloured dot. The color code of each dot (station)
5  refers to the maximum BrO vmr reported in literature (Arrival Heights: Kreher et al., 1997; Neumayer: Frieß et al., 2004; Dumont dÚrville:
Grilli et al., 2013; Halley: Roscoe et al., 2014; Dome-C: Frey et al., 2015; Marambio and Belgrano: this work). Note that only the present
study provides contemporary observations from different sites. Further details are provided on Table 1.





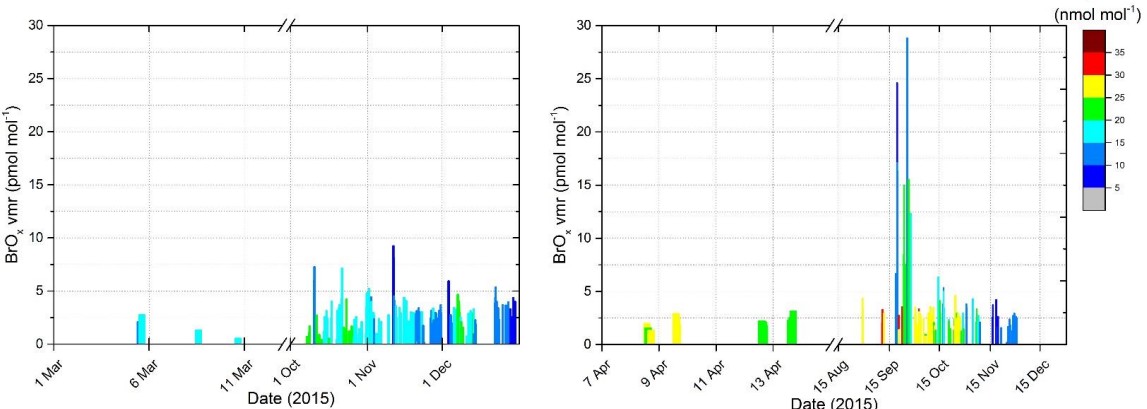

**Figure 12: Reactive bromine in the low troposphere of Belgrano (left) and Marambio (right) under different O₃ regimes.** The vertical scale depicts the BrOₓ (Br+BrO) at each station and the color code refers to the collocated observed O₃ vmr. Note that the vertical scale and the color code apply to both figures. Only observations performed under low wind conditions (< 6 m s⁻¹) are included.

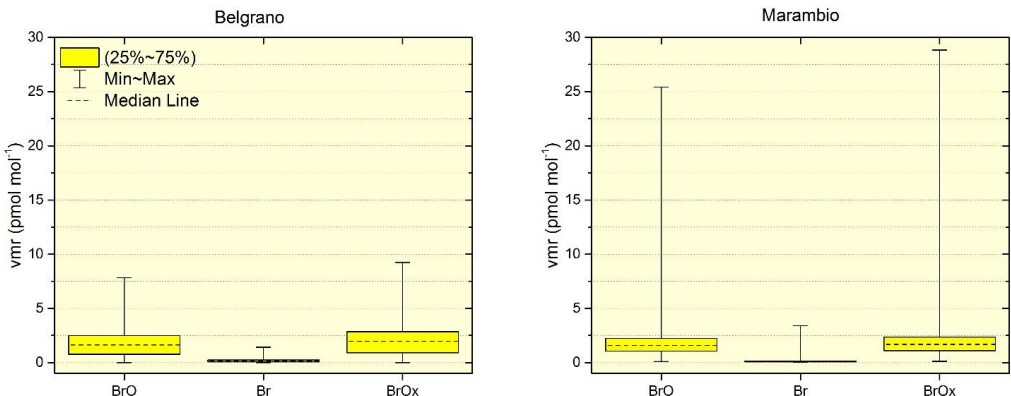

**Figure 13: Statistical analysis of the reactive bromine and its partitioning estimated at Belgrano (left) and at Marambio (right) during the sunlit period of 2015.** The vertical scale, which is the same in both plots, indicates the range of mixing ratios of BrO, Br and BrOₓ at both stations. The legend applies to both figures, where the whiskers display the range of the maximum and minimum vmr, the boxes in dark yellow provide the vmr ranges of 25-75 % of the data, while dashed lines depict the median vmr.





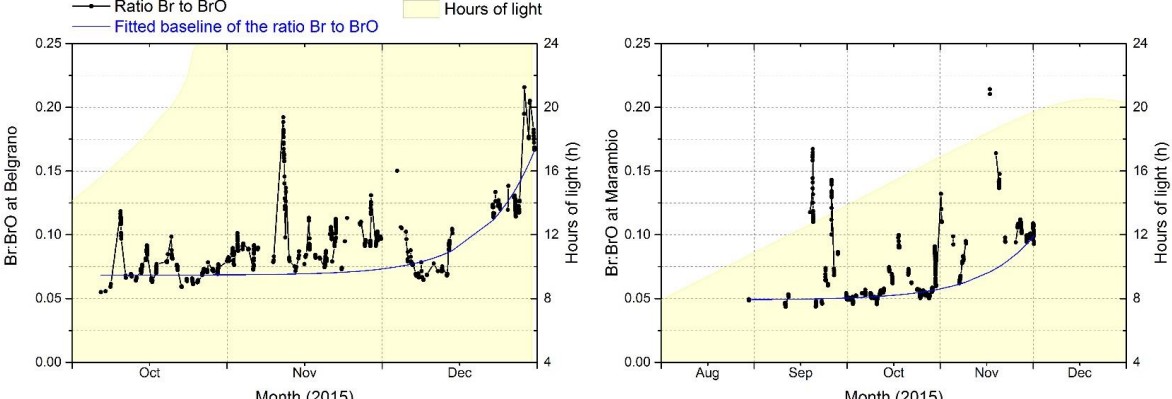

**Figure 14: Variability of the ratio Br to BrO after the polar sunrise at Belgrano (left) and Marambio (right).** The left axes refer to the Br to BrO ratio (same scales on both plots) with the estimated ratios shown in black and the fitted baseline in blue. The right axes on both
5    plots refer to the hours of light at each station. The horizontal scales indicate the time period. Note that only data observed under low wind conditions are considered.