# Peer review of "Reactive bromine in the low troposphere of Antarctica. Estimations at two research sites."

_Atmospheric Chemistry and Physics, 2018_

## Referee Comment (RC1) · Anonymous Referee #1 · 5 Mar 2018

The paper by Prados-Roman et al. presents observations of reactive bromine in the lower troposphere at two Antarctic research sites from where no data was available until now. This adds to only a handful of Antarctic sites where tropospheric bromine chemistry has been investigated so far. Based on MAX-DOAS measurements, they retrieve vertical profiles of aerosol extinction and the BrO radical with an optimal estimation algorithm and complement the results with meteorological observations and surface ozone measurements. Lastly, the amount of reactive bromine at both sites is estimated.

[Figure]

**1 General comments**

While the observations presented in the paper surely have the potential to add interesting and important insights to our understanding of tropospheric bromine chemistry in polar regions, in particular with the observations of a high activity at Marambio, the site on the Antarctic peninsula, I recommend substantial additions to the documentation of the profile retrieval analysis followed by a critical review of its interpretation and the conclusions of the paper before publication.

**1.1 Documentation of the profile retrieval process and interpretation of the resulting data**

Profile retrieval based on optimal estimation (e.g. Rodgers 2000) as it is used in this publication, is a method to tackle ill-posed inversion problems (in this case the conversion of differential slant columns from MAX-DOAS measurements to vertical profiles of aerosol extinction and trace gases). The problem is ill-posed because the observations alone do not contain enough information to fully determine the state of the atmosphere. Therefore, a-priori information (based on independent knowledge e.g. a climatology) is required for the inversion process. The result of such an inversion is an optimally estimated new state of the atmosphere based on the information contained in the measurements and the a-priori. A meaningful change/update of the a-priori state is only possible, where the instrument is sensitive enough - for the application in this paper both spatially (vertically) and in terms of measurement precision.

Regarding the documentation of the inversion process presented in this paper, the following information should to be added and discussed:

To allow a transparent assessment of the presented profile inversion and to ensure reproducibility and comparability with similar observational data, quantitative information about the quality of the dSCD data should be provided (e.g. a statistic of the DOAS fit

error that was used in the inversion as mentioned on page 5 line 23).

To allow an assessment of the aforementioned contribution of the measured data to the retrieved profiles (vs a-priori information) and to judge the vertical sensitivity of the measurements, representative averaging kernels should be presented and absolutely have to be discussed. One of the conclusions of the paper (absence of BrO above 2km) is based on the claim that profiles 'in the first 6km were measured' (page 8 line 4). The vertical sensitivity of inverted MAX-DOAS measurements implied with this is much higher than in publications e.g. by Roscoe et al. (2014), Peterson et al. (2015) or Franco et al. (2015) where comparable sequences of elevation angles and optimal estimation methods were used. The higher vertical sensitivity claimed in this study should be well substantiated or interpretation and conclusions changed. This potentially requires changes to the data presentation as well.

E.g. in the plots of vertical profiles of the entire data set (figure 8) and selected days (figure 9), the presented profiles should be limited to a vertical extent that is based on this analysis and discussion of averaging kernels and the vertical sensitivity. The use of colour map/contour plots should be limited to qualitative discussions (if used at all) as they suggest a higher information content (in terms of vertical resolution - especially when a smoothing between the retrieved layers is used) than can be expected from inverted MAX-DOAS dSCDs and hence could be misleading for readers not familiar with the details of profile retrievals. For quantitative analysis or discussions of the question of elevated layers and export to the free troposphere, profiles based on the information content of the retrieval should be generated. Integrating all layers in the lowermost 2km, as was already done in the analysis in this publication is one, albeit quite conservative solution here. Other publications have produced profiles with vertical layers based on the degrees of freedom (e.g. Roscoe et al. 2014) consisting of two to three layers.

For the comparison with ground-based measurements such as the ozone time series presented here, as well as the estimation of BrOx, again the averaging kernels of the

lowermost layers should be considered and all layers with a non-negligible influence on the surface layer results (or at least all layers covered by the width of the averaging kernel corresponding to the lowermost layer) should be integrated rather than just selecting the lowermost layer.

See also suggestions in the Specific Comments section.

**1.2 Processing of ancillary data for the interpretation of BrO observations**

The ancillary data provided in this study, surface ozone and meteorological observations, are presented in a quite general way and do not provide very specific information that could help the interpretation of the BrO observations. While a description of the general metrological conditions of the two sites and the differences between them (as in table 3) are important, the data provided somewhat lacks a real connection to the observed periods of elevated BrO/ODEs. It would be desirable to have meteorological times series filtered to reflect the periods of profile data presented. For example, it would add important context and improve the quality of this study, if wind directions could be filtered for the periods with elevated BrO as this could provide first insights about the origin of the observed air masses. A correlation of surface ozone and BrO mixing ratios in the lowermost layers of the retrieved profiles could help determine, whether an air mass already depleted in ozone/enriched in BrO was observed or the chemistry happened locally.

**2 Specific comments**

page 1 line30: What is a heterogeneous increase? (Exponential) increase by heterogeneous reactions/chemistry?

p.3 l.11: Hüneke et al. focuses on upper troposphere lower stratosphere, better example of airborne measurements e.g. Peterson et al. (2017) already cited on p.4 l.13

p.4 l.12: Reference missing Bobrowsky(?)/Bobrowski et. al. 2003

p.5 l.14f: What does the goal of long term observations entail for parameter selection in detail?

p.5 l.23: dSCD errors: Please provide statistics about these errors (mean and std). How many sequences of elevation angles were used for one profile? What is hence the temporal resolution?

p.5 l.23f: a-priori errors: Please elaborate briefly on the idea behind this approach and what $\beta$ is. This approach means that the statistics of the inversion is no longer Bayesian (in contrast to Rogers 2000). This should be underlined. Was this used for the a-priori of both AECs and BrO and why? Clémer et al. (2010) would be a better citation for the details of this approach.

p.5 l.25: Why is the correlation length different for aerosols and trace gases?

p.5 l.26: A brief explanation what these errors are would improve clarity here. If the combined error is used as 'inversion error' later on (p.8 l.41f/9 l.1), this should be defined here

p.5 l.30: Albedo of 0.8: This value is too low for the UV spectral range. A value of about 0.98 would be more appropriate between 300-400 nm (see Grenfell et al. 1994)

p.5 l.31: AOD limit: How does this limit work exactly? Are retrievals with AOD larger than 5 filtered out afterwards or is there an internal limit? What does that mean for meteorological conditions namely cloud cover? What cloud cover conditions are filtered out by this?

p.5 l.32: What are typical DOFs of the data set?

p.5 l.39: What is meant by 'lower differences'? By definition, the a-priori should be independent information (e.g. from a climatology). This sounds like the O4 dSCDs

were used to optimise the a-priori scale heights prior to using them as information independent of the measurements. What could be an explanation for the difference and the very high SH of 2km?

p.6 l.36: As mentioned in section 2.2.1...: This is not discussed in 2.2.1 at all (but should be - based on averaging kernels)

p.7 l.12f: Please make clear which station is referred to. (There is no Polar night at Marambio!) What is meant by 'BrO levels were undetectable just before...and immediately after... '? The data set provided here reports no MAX-DOAS observations within one month from both of these points in time. Does that mean no data or no BrO observations? (compare Fig. 3 and 4)

p.7 l.13: What is meant by 'the magnitude and variability of the BrO maximums direct the difference'? What is the maximum referred to? A daily maximum?

p.8 l.4: the VMRs were not 'measured' rather estimated.. The claim of sensitivity up to 6km altitude should be substantiated (see General comments).

p.8 l.5: Absence of elevated layers: If this actually can be inferred should be reviewed after assessment of the vertical sensitivity (see General comments)

p.8 l.23ff: Discussion of BrO vmr vertical profiles. These profiles should be generated based on information content/the analysis of the averaging kernels (see General comments). The presentation chosen here -without error bars and with a smoothed profile rather than visualising the single layers of the retrieval is misleading for readers not familiar with the details of MAX-DOAS profile inversions. In such a figure, the (typical) a-priori and its errors should be presented as well.

p.8 l.41: Whether or not the detection of BrO above 2km can be discussed with the presented data set strongly depends on vertical sensitivity.

p.8 l.41f/p.9 l.1: The inversion error was not clearly defined (see comment p.5 l.25)

p.9 l.3.: BrO in the free troposphere: The absence of BrO and hence an upper limit for BrO in the free troposphere should only be concluded for altitudes where the MAX-DOAS observations are sensitive (based on averaging kernels, see general comments). Based on the data presented, in my opinion, the absence of BrO (above the stated detection limit) in the free troposphere can only be concluded, if one assumes an upper limit of the boundary layer of 1500m or higher because the retrieval quite consistently seems to yield mixing ratios of BrO of at least 1.5 pmol/mol above 1000m altitude (Figure 8 or Belgrano example from 29th October in figure 9). I am not convinced that an altitude of 2000m for the top of the boundary layer is a very regular event in polar regions. Indeed, the two cited publications show e.g. a maximum depth of the convective boundary layer of about 300m for Halley station (King et al. 2006) and give altitudes for the humidity inversion at Marambio of 700m-1300m throughout the year with values of 1000m in spring (Nygard et al 2013). The data as presented here, shows considerable mixing ratios above the detection limit between 700 and 1300m. If the vertical sensitivity at these altitudes is sufficient, discussions about the export of BrO to the free troposphere or the absence thereof should be based on information about the typical depth of the boundary layer at the respective times of the year at the two locations (e.g. the radiosonde data at Belgrano used in the retrieval or -if available- the radiosondes at Marambio used in Nygard et al. 2013) rather than on the stated (quite wide) range of possible altitudes for the top of the boundary layer of 100m to 2km.

p.9 l.24: Estimation of BrOx: Please make the assumptions going into this clearer. I do not understand at all how the observational data feeds into this estimation. Is this an estimation at noon? Was the daily average of BrO and O3 used or the maximum? The rate of BrO photolysis assumed here should be stated. This rate strongly depends on the actinic flux which in turn depends on visibility conditions/cloud cover. How was this treated exactly? What influence does the filtering of total AODs above 5 mean for this? HO2 has a very pronounced daily cycle as well. What justifies the selected, fixed mixing ratio?

p.9 l.31: What exactly is shown in this figure? Daily averages? maximum values?

p.9 l.40: What is meant by 'troposphere reactivity'? oxidation capacity?

p.10 l.13: This seems a bit circular to me. The O3 measurements were used to calculate BrOx which then is used to estimate the O3 loss rate? Please elaborate your reasoning behind this.

p.11 l.4: I would add 'at two (new) sites' after 'inorganic bromine'. The sentence as it is sounds a bit as if the results apply to the entire Antarctic troposphere while the observations are already quite different for the to sites (as mentioned by the authors later on)

Table1: measurement period column: Please only use months and days here. The use of the term 'season' in an Antartic context is misleading as it could also mean three (summer) seasons. The indicated periods from this study also should rather be 4.5 months than '3 seasons' since the periods from the other publications also only state periods of reported observations and not the entire time when the instruments were deployed.

Table3: days with snowfall: Is this only precipitation (excluding blowing snow)? Information about days with blowing snow would be interesting as well - if available.

Figures3+4: Please provide errors and detection limits in these plots

Figure5: This data is not very helpful. The averaging window could be increased to show wind regimes in different seasons. Alternatively, a histogram of wind speeds would provide more information.

Figure6: Additionally, data filtered for the days presented in the profile data would be desirable

Figure7: It would be nice to have the reported periods of MAX-DOAS observations marked in these plots.

Figure8: These plots are quite small. A blow up or separation of the two periods would benefit the information conveyed by them. The vertical axis should be adapted to the updated vertical sensitivity. Plotting vmr boxes with the size of the retrieval grid (pixels) or aggregated profiles based on averaging kernels rather than using the linear(?) smoothing in the colour map plots would make the nature of the retrieval process and the resulting data clearer. It would also be good to clearly mark periods where data is missing or was filtered out (e.g. based on the AOD limit). For example, it is not clear to me if the periods in the BrO profiles from Marambio in December are missing data or just no BrO at all for half a month.

Figure9: See remarks to Figure 8 regarding smoothing and axis. For the example day from Marambio on September 25th, the data selection should be reviewed. On that day, the SZA limit stated on page 4 of 75 degrees is reached already at 19:25 while the plot shows data until 20:00

Figure10: Dots to indicate the values in the different levels of the retrieval, error bars and a-priori profiles with the respective errors should be added here

Figure11: The location of Belgrano and Marambio is the only new information in this figure. Maximum values are already provided in table 1. As these values are from different years, plotting them in such a manner could be misleading.

Figure12: What is meant by 'depicts the BrOx [...] at each station' Please make clear what is plotted here. Daily averages or maximum values?

**3 Technical comments**

page 2 line 27: add s: high amounts

p.4 l.3: remove s: aerosol extinction

p.4 l.31: installed in

p.5 l.14: consists of

p.5 l.11: of a two-step approach

p.5 l.32: taken into consideration

p.6 l.19: installed at the site

p.6 l.32: Herein,

p.8 l.33: considerably stronger

p.10 l.13: this simplified scheme

p.10 l.25: conclusions from

p.11 l.9: performed at the two sites

Table1 - Caption title: observations of tropospheric BrO made in Antarctica

**4 References**

Clémer, K., Van Roozendael, M., Fayt, C., Hendrick, F., Hermans, C., Pinardi, G., Spurr, R., Wang, P., and De Mazière, M.: Multiple wavelength retrieval of tropospheric aerosol optical properties from MAXDOAS measurements in Beijing, Atmos. Meas. Tech., 3, 863-878, https://doi.org/10.5194/amt-3-863-2010, 2010.

Franco, B., Hendrick, F., Van Roozendael, M., Müller, J.-F, Stavrakou, T., Marais, E. A., Bovy, B., Bader, W., Fayt, C., Hermans, C., Lejeune, B., Pinardi, G., Servais, C., and Mahieu, E.: Retrievals of formaldehyde from ground-based FTIR and MAX-DOAS observations at the Jungfraujoch station and comparisons with GEOS-Chem and IMAGES model simulations, Atmos. Meas. Tech., 8, 1733-1756,

https://doi.org/10.5194/amt-8-1733-2015, 2015.

Grenfell, T. C., S. G. Warren, and M. C. Mullen (1994). "Refelection of solar radiation by the Antarctic snow surface at ultraviolet, visible, and near-infrared wavelengths". In: Journal of Geophysical Research 99.D9, pp. 18, 669–18, 684. doi: 10.1029/94JD01484.

Peterson, P. K., Simpson, W. R., Pratt, K. A., Shepson, P. B., Frieß, U., Zielcke, J., Platt, U., Walsh, S. J., and Nghiem, S. V.: Dependence of the vertical distribution of bromine monoxide in the lower troposphere on meteorological factors such as wind speed and stability, Atmos. Chem. Phys., 15, 2119-2137, https://doi.org/10.5194/acp-15-2119-2015, 2015.

Peterson, P. K., Pöhler, D., Sihler, H., Zielcke, J., General, S., Frieß, U., Platt, U., Simpson, W. R., Nghiem, S. V., Shepson, P. B., Stirm, B. H., Dhaniyala, S., Wagner, T., Caulton, D. R., Fuentes, J. D., and Pratt, K. A.: Observations of bromine monoxide transport in the Arctic sustained on aerosol particles, Atmos. Chem. Phys., 17, 7567-7579, https://doi.org/10.5194/acp-17-7567-2017, 2017.

H.K. Roscoe, N. Brough, A.E. Jones, F. Wittrock, A. Richter, M. Van Roozendael, F. Hendrick, Characterisation of vertical BrO distribution during events of enhanced tropospheric BrO in Antarctica, from combined remote and in-situ measurements, Journal of Quantitative Spectroscopy and Radiative Transfer, Volume 138, 2014, Pages 70-81, ISSN 0022-4073, https://doi.org/10.1016/j.jqsrt.2014.01.026.

---

## Referee Comment (RC2) · Anonymous Referee #2 · 16 Mar 2018

**General Comments**

This manuscript gives an overview of the retrieval of BrO vertical profiles and column densities from ground-based MAX-DOAS measurements at 2 locations on the Antarctic coast. They examine the impacts of meteorology, aerosols, and ozone on the observed BrO, as well as doing some simple modeling to determine ozone lifetimes due to $BrO_x$ at each site. They find differences in BrO between the two sites and suggest these differences could be linked to differing sea ice condions between the two sites. They also find that BrO enhancements occur under low wind speed conditions and that blowing snow is not needed for substantial enhancements, but rather that surface emissions and vertical mixing can be responsible for the observed enhancements. This work is

well presented, within the scope of ACP, and merits publication after addressing some minor issues with the MAX-DOAS interpretation and analysis that I discuss below.

**Specific Comments**

MAX-DOAS analysis

The authors claim that BrO is not present in significant amounts above 2 km based on their ground-based measurements, where they retrieve BrO vertical profiles from 0-6 km. I am skeptical that ground-based MAX-DOAS measurements can be used to make this claim. For what it is worth, I am skeptical that the prior studies cited could actually observe BrO at those altitudes as well. The information content outside of the lowest elevation angle measurements simply isn't high enough. The authors should present averaging kernels showing that the measurements are sensitive to changes in BrO above 2 km if they are going to make this claim. I also think the presented vertical profiles should also be limited to 2 km unless the averaging kernels show that a higher altitude is merited.

Sea ice conditions between the two sites

I think the author's points about needing to examine the sea ice conditions at both sites and the heterogeneity being potentially linked to sea ice differences is a good one. However, I think simply describing the sea ice around Marambio as seasonal without providing further detail is potentially misleading, as the ice toward the outside edge of the sea ice in the Antarctic is often the oldest sea ice (excluding the "permanently" sea iced sections surrounding Belgrano) (Nghiem et al., 2016). This older sea ice is likely lower salinity than the newer sea ice regions closer to the coast. These differences

may impact the overlying snowpack, which is the likely source of the reactive bromine. Of course the proximity of this older ice to open water may also lead to enhanced snow salinity due to sea spray aerosol deposition (e.g. May et al., 2016). In any case, I'd like to see the authors add a more detailed discussion of the sea ice conditions at the two sites.

Page 1, Line 41

This sentence should have references for these impacts of atmospheric halogens.

Page 5, Line 32

What percentage of the retrievals has a DOF larger than 1?

Page 5, Line 40

A summary of the degrees of freedom for the BrO retrievals should be presented here as well.

Page 6, Line 39

Can you state the differences in AOD between the two sites more quantitatively?

Page 7, line 11

$0.8 \times 10^{13}$ molec cm$^{-2}$ isn't a range as presented. Please clarify, is this a standard deviation?

Suggested Figure Modifications

1. Figures 3,4: I don't really think it is necessary to shade regions without data. It gives the figure a cluttered look.

2. I think just showing Fig. 6 is sufficient, and the timeseries of wind speed (Fig. 5) isn't really needed.

3. Fig. 7: Consider plotting both ozone series on the same panel so one can clearly see the differences between the two sites.

4. Fig. 8,9,10: As I suggest above, the portion above 2 km should be cut unless you can demonstrate that your retrieval is sensitive to the true atmospheric state at higher altitudes.

**References**

May, N. W., Quinn, P. K., McNamara, S. M., and Pratt, K. A.: Multiyear study of the dependence of sea salt aerosol on wind speed and sea ice conditions in the coastal Arctic, Journal of Geophysical Research: Atmospheres, 121, 9208–9219, doi:10.1002/2016JD025273, http://doi.wiley.com/10.1002/2016JD025273, 2016.

Nghiem, S. V., Rigor, I. G., Clemente-Colón, P., Neumann, G., and Li, P. P.: Geophysical constraints on the Antarctic sea ice cover, Remote Sensing of Environment, 181, 281–292, doi:10.1016/j.rse.2016.04.005, http://www.sciencedirect.com/science/article/pii/S0034425716301481, 2016.

---

## Author Comment (AC1) · 24 May 2018

The authors would like to thank Referee #1 for the detailed reading of our work and his/her insightful comments. Below, we address those comments in a point-by-point basis. Note that, in addition to the draft, we now include a Supplementary Material where the reader is kindly referred to if he/she is interested on particular details on the MAX-DOAS and inversion scheme. Through this Supplementary Material, we not only keep the simplicity of the draft but also provide further insights on the more technical information behind our work. Overall, we consider that our work has improved after both referees comments and suggestions and we are truly grateful for that.

In the following, the Referee Comments are in black and Authors Comments in blue. References to pages and lines of the draft are indicated with "P" and line "L", respectively.

**1. General comments**

**1.1. Documentation of profile inversion and interpretation of the resulting data**

Regarding the documentation of the inversion process presented in this paper, the following information should to be added and discussed:

To allow a transparent assessment of the presented profile inversion and to ensure reproducibility and comparability with similar observational data, quantitative information about the quality of the dSCD data should be provided (e.g. a statistic of the DOAS fit error that was used in the inversion as mentioned on page 5 line 23).

For the sake of simplicity for the reader, these DOAS fit statistics have now been included in the Supplementary Material, Sect. 1 (and thus indicated in the draft P5, L23).

To allow an assessment of the aforementioned contribution of the measured data to the retrieved profiles (vs a-priori information) and to judge the vertical sensitivity of the measurements, representative averaging kernels should be presented and absolutely have to be discussed.

One of the conclusions of the paper (absence of BrO above 2km) is based on the claim that profiles 'in the first 6km were measured' (page 8 line 4). The vertical sensitivity of inverted MAX-DOAS measurements implied with this is much higher than in publications e.g. by Roscoe et al. (2014), Peterson et al. (2015) or Franco et al. (2015) where comparable sequences of elevation angles and optimal estimation methods were used. The higher vertical sensitivity claimed in this study should be well substantiated or interpretation and conclusions changed. This potentially requires changes to the data presentation as well.

E.g. in the plots of vertical profiles of the entire data set (figure 8) and selected days (figure 9), the presented profiles should be limited to a vertical extent that is based on this analysis and discussion of averaging kernels and the vertical sensitivity. The use of colour map/contour plots should be limited to qualitative discussions (if used at all) as they suggest a higher information content (in terms of vertical resolution – especially when a smoothing between the retrieved layers is used) than can be expected from inverted MAX-DOAS dSCDs and hence could be misleading for readers not familiar with the details of profile retrievals. For quantitative analysis or discussions of the question of elevated layers and export to the free troposphere, profiles based on the information content of the retrieval should be generated. Integrating all layers in the lowermost 2km, as was already done in the analysis in this publication is one, albeit quite conservative solution here. Other publications have produced profiles with vertical layers based on the degrees of freedom (e.g. Roscoe et al. 2014) consisting of two to three layers.

Typical Averaging kernels obtained in this work shows that, depending on the extinction condition (scattering and/or absorption) our method is sensible up to an altitude of about approximately 4 km (see Supplementary Material, Sect. 2).

To find out what would be observed by our MAX-DOAS instruments if a layer of aerosols and/or BrO would be placed at high altitudes, we have performed some sensitivity tests (see Supplementary Material, Sect. 3). Results of these simulated scenarios indicate a vertical sensitivity of up 4 km for aerosols and 2.5-3 km for BrO. Thus, figures in this work have been modified to represent our altitude sensitivity. Note that (P1 L29-30 of the initial draft) "and undetectable values above 2 km at both sites" has been deleted. See also responses below.

For the comparison with ground-based measurements such as the ozone time series presented here, as well as the estimation of BrOx, again the averaging kernels of the lowermost layers should be considered and all layers with a non-negligible influence on the surface layer results (or at least all layers covered by the width of the averaging kernel corresponding to the lowermost layer) should be integrated rather than just selecting the lowermost layer.
See also suggestions in the Specific Comments section.

Certainly, each technique has its own benefits but also its own limitations. Thus, the comparison of different techniques usually comes with limitations as well. Indeed, although most of the information retrieved at the surface (i.e., lower layer) with the DOAS observations are mainly influenced by the state of the atmosphere at that layer, in fact they are rarely 100% independent of the state of the atmosphere some the layer(s) just above (e.g., Supplementary Material, Sect. 2). Moreover, the DOAS observations at the instrument's altitude (surface in this case) in fact provide information not only at the instrument location but they integrate information of the atmosphere contained up to several kilometers away of the instrument (depending on the scattering conditions) in the horizontal field of view of the instrument. On the contrary, the ozone observations facilitated in this work do provide information of the amount of O3 present at the very exact instrument's location (in the horizontal and in the vertical) and, unfortunately, there are not in situ measured O3 profiles at both stations on hourly basis. Indeed, any mixture of data obtained by remote and by in situ techniques should be taken with caution. Note that, proceeding as Referee #1 suggests with the DOAS data will only overcome partially the vertical information issue when using in situ and remote sensed data (partial due to the lack of vertical information of the O3 measurements anyway). In addition, there is nothing we can do with to overcome the in situ "punctual" information in the horizontal part compared to the DOAS data. However, given the sort of data (and related limitations) we have, what we can do is acknowledge this sort of limitations in the draft. This is now done on P11, L8-12:
*"Note that the estimations provided herein are limited by the information content inherent in the in situ technique measuring near-surface ozone (i.e., information at the exact instrument's location) compared to the DOAS data which integrates information several kilometers away of the instrument (depending on scattering conditions) in the horizontal field of view and also in the vertical (see retrieved averaging kernels in the Supplementary Material)."*.
Also, the sentence *"Further studies including different sources and sinks of bromine radicals in the Antarctic environment would be needed to confirm these numbers…"* (P10, L22 of the initial draft) is now rephrased as *"Further studies would be needed to confirm these numbers, including investigations on different sources and sinks of bromine…"* (P11, L12-13).
Worth noticing is that, as it was mentioned already on the draft, the calculations presented on Sec. 3.3 are *"approximations"* (P10, L25 of initial draft) and *"simplified scheme"* (P10, L13 of initial draft) and therefore we could only *"estimate"* BrOx (e.g., P9, L19 and L41 or P10 L4 of initial draft). Note that, however, the different approximations involved on our BrOx estimation seem not far from reality at least at Marambio (P10, L36-38 of initial draft). Moreover, we clearly stated on the draft that *"Further studies including different sources and sinks of bromine radicals in the Antarctic environment would be needed to confirm these numbers"* (P10, L22 of the initial

draft) and that *"Additional investigations on the variability and geographical distribution of the bromine source gases throughout the year are suggested to address the bromine pathways in the Antarctic troposphere and their consequences"* (P10, L11-13 of the initial draft) while *"dedicated investigations combining models and collocated observations of e.g. halogenated substances (not only BrO), organic compounds, DMS, NOx, HOx, particles and sea ice properties at different stations, could assist a thorough study of the bromine sources and pathways at Antarctica, their geographical distribution and their projections under a changing environment"* (P11, L31-34 of the initial draft).

**1.2. Processing of ancillary data for interpretation of BrO observations**

The ancillary data provided in this study, surface ozone and meteorological observations, are presented in a quite general way and do not provide very specific information that could help the interpretation of the BrO observations. While a description of the general metrological conditions of the two sites and the differences between them (as in table 3) are important, the data provided somewhat lacks a real connection to the observed periods of elevated BrO/ODEs. It would be desirable to have meteorological times series filtered to reflect the periods of profile data presented. For example, it would add important context and improve the quality of this study, if wind directions could be filtered for the periods with elevated BrO as this could provide first insights about the origin of the observed air masses. A correlation of surface ozone and BrO mixing ratios in the lowermost layers of the retrieved profiles could help determine, whether an air mass already depleted in ozone/enriched in BrO was observed or the chemistry happened locally.

The presented work aims at assessing the presence and the seasonal variation of BrOx at the two sites of Antarctica during the sunlit period of 2015. Although surely worth investigating in a forthcoming work, the particular analysis of a given ODE is out of the scope of this paper. Note that such halogens-ODE exercise in polar regions has already being nicely performed in rather recent works (e.g. Thompson et al. 2015 or Halfacre et al. 2014).

Aiming at the aforementioned 2015 description of our work, an overview of the 2015 weather conditions is summarized in Table 3.  However, since we understand the Referee's point, the figure of the wind roses (Fig. 5 of the new draft) is now adapted to show the wind directions depending on the wind speeds regimes observed and also referred to throughout the work (low or < 6 m s$^{-1}$, medium or 6 – 12 m s$^{-1}$ and high wind speeds or > 12 m s$^{-1}$). Worth noticing is that the 2015 summary presented in the wind rose (Fig. 5, Marambio) is in agreement with the recently publish work of Asmi et al. (2018) focused on site of Marambio and comprising the 2013-2015 period. This agreement is now mentioned in the new draft (P7 L40) and that reference added to the bibliography.

Thus, in the new draft, the paragraph (P7, L23-26 of initial draft)
*"Regarding the wind measurements, the speed observed at both stations is shown in Fig. 5, while the wind roses are given in Fig. 6. Observations indicate that, although the higher gusts of wind were quite similar at both stations (~120 km h-1), the mean wind speed at Marambio was in general 50% higher than in Belgrano"*
is changed to
*"Regarding the wind measurements, the wind rose of the 2015 measurements at each station is shown in Fig. 5 for low, medium and high wind speeds. Note that the information reported for 2015 at Marambio is consistent with the recent publication of Asmi et al. (2018) referred to the 2013-2015 period. The 2015 observations at both stations indicate that, although the higher gusts of wind were quite similar at both stations (~34 m s-1), the median wind speed at Marambio (7.2 m s-1) was in general 50% higher than in Belgrano."* (P7 L38-40 to P8 L1).

Worth mentioning is also that, as stated on P9 L28 of the initial draft, the BrOx study presented in Sect. 3.3 is based only on data during low wind conditions (< 6 m s-1). Since, indeed, some exemplary data (under wind speed < 10 m s$^{-1}$, P8, L10 of the initial draft) of BrO are presented in this work (e.g., Fig. 8 of the new draft), following the referee suggestion, details regarding the wind conditions on the specific days is now added in the new version of the draft (first paragraph of Sec. 3.2).

**2. Specific comments**

P1 L30: What is a heterogeneous increase? (Exponential) increase by heterogeneous reactions/chemistry?
In the new draft "geographical" is added on P1 L30 and P12 L13.

P3 L11: Hüneke et al. focuses on upper troposphere lower stratosphere, better example of airborne measurements e.g. Peterson et al. (2017) already cited on p.4 l.13
The work of Hüneke et al. (2017) is based on aircraft-borne observation during different sections of diverse flights. Within that work, the authors include a dive into the Antarctic troposphere (65° S, 21° E) down to an altitude of 4 km. On our work, the study of Hüneke et al (2017) is referred to as one of the few studies investigating "the presence of tropospheric BrO in the Antarctic region" (P3 L8-11 of the initial draft). We agree with the referee and the work of Peterson et al (2017) is an interesting study already referred to within our draft. However, the work of Peterson et al (2017) refers to Arctic observations. Since that part of our draft focuses only on Antarctic observations, we consider Hüneke et al (2017) should remain as a relevant reference with regard BrO observations in the Antarctic atmosphere.

P4 L12: Reference missing Bobrowsky(?)/Bobrowski et. al. 2003
The new version of the draft already includes the missing reference in the bibliography.

P5 L14: What does the goal of long term observations entail for parameter selection in detail?
We mean that the parameterization of the radiative transfer model is based on general properties of the Antarctic atmosphere and not particularized for given events (e.g., blowing snow, ODEs…). Indeed, when a study of this kind is limited to a few days, the choice of the parameters determining the vertical retrieval can be performed taking into account the particular conditions of each of those days. However, when long-term observations are used, including a large number of days, the selection of the parameters has to be adequate for a large number of situations too. Thus, the parameterization does not aim not be "optimal" for a particular day but for the whole period.

P5 L23: dSCD errors: Please provide statistics about these errors (mean and std). How many sequences of elevation angles were used for one profile? What is hence the temporal resolution?
As mentioned above, statistics of the dSCD errors are now included in the Supplementary Material. The set of elevation angles was provided in the draft on Table 2 and the time resolution on P4 L29 of the initial draft (i.e., "every 15 minutes").

P5 L23f: a-priori errors: Please elaborate briefly on the idea behind this approach and what _ is. This approach means that the statistics of the inversion is no longer Bayesian (in contrast to Rogers 2000). This should be underlined. Was this used for the a-priori of both AECs and BrO and why? Clémer et al. (2010) would be a better citation for the details of this approach.

We agree with the referee, this point should be clarified and explained in more detail in the manuscript. Reference of Franco et al., (2015) has been replaced by Clémer et al. (2010) and deleted from bibliography and additional details on the profile retrieval are now provided with the following text (P5 L23-32):

*"The diagonal elements of the a priori covariance matrix in the inversion was calculated as 100 % of the a priori profile for BrO and based on Clémer et al. (2010) for the aerosol extinction retrieval. Note that, given the large variability of the visibility conditions at Antarctica, the true aerosol profile can strongly differ from the a priori profile. In order to allow the extinction profile retrieval to capture these variations, we followed the method described in Clémer et al. (2010). In this method, the diagonal element of Sa closest to the surface (i.e., Sa (1,1)) is set equal to the squared of a scaling factor (ß) times the maximum partial AOD of the extinction profile obtained in the precedent iteration. In this study, ß has been set to 1. The other diagonal elements decrease linearly with altitude down to 20% of Sa (1,1). Note that, despite not being a statistically Bayesian method, it allows the profiles corresponding to large AOD to differ significantly from the a priori, while the profiles with smaller AOD present lower variations from the a priori."*

P5 L25: Why is the correlation length different for aerosols and trace gases?
The correlation length indicates how the particles or gases of one layer interact with their neighbors of the adjacent layers. Prior the study, several tests were performed for aerosols and BrO profile retrievals using three values for the correlation length: 100, 200 and 300 m which are standard for MAX-DOAS retrievals (e.g., Clémer et al. 2010; Frieß et al., 2011). A better agreement between measured and simulated O4 and BrO DSCDs was found when a correlation length of 100 m and 300 m was used for aerosol extinction and BrO profiles, respectively.

P5 L26: A brief explanation what these errors are would improve clarity here. If the combined error is used as 'inversion error' later on (p.8 l.41f/9 l.1), this should be defined here
To clarify this point, the sentence *"the error of the retrieved profiles provided in this work contains the measurement error and the smoothing error of the retrieval"* (P5, L25-26 of the initial draft) is rephrased to *"the error of the retrieved profiles provided in this work contains the measurement error (experimental dSCDs error) and the smoothing error of the retrieval. The last takes into account that the retrieval is an estimate of the true profile smoothed by the averaging kernel functions."* (P5 L33-36)

P5 L30: Albedo of 0.8: This value is too low for the UV spectral range. A value of about 0.98 would be more appropriate between 300-400 nm (see Grenfell et al. 1994)
During 2007, in the context of the Arctic Study of Tropospheric Aerosol, Clouds and Radiation (ASTAR) campaign, the sea ice albedo was measured in the Arctic from 310 to 930 nm using a Spectral Modular Airborne Radiation measurement sysTem (SMART-Albedometer). This work is summarized in the PhD thesis of André Ehrlich. Results indicate that in the UV-A spectral range, the sea ice albedo varies from 0.79 up to nearly 1 (see also Prados-Roman et al., 2011). In particular, at 360 nm (wavelength of our interest), the measured sea ice albedo during ASTAR was ~0.82 (Fig. 5.1 of Ehrlich, 2009). This value is in good agreement with the value used on our study and thus it is now mentioned in the new draft (P5, L40-41) and the reference of Ehrlich (2009) added to the bibliography. Furthermore, previously to the work here presented, diverse tests were performed for aerosol extinction and BrO retrievals using surface albedos of 0.8 and 0.9. A better agreement between measured and simulated O4 and BrO DSCDs was obtained when an albedo of 0.8 was used (P5 L30 of the initial draft).

P5 L31: AOD limit: How does this limit work exactly? Are retrievals with AOD larger than 5 filtered out afterwards or is there an internal limit? What does that mean for meteorological conditions namely cloud cover? What cloud cover conditions are filtered out by this?

Note that, by mistake, we wrote "5" instead of "0.5" as the upper limit of AOD (i.e., observations with AOD>0.5 were considered as invalid for profile inversion of BrO). This is corrected in the new draft (P5, L42). Filtering the data in this way, we avoid including data affected by high aerosol loads that could complicate the interpretation of the BrO results. In those situations (high AOD), scattering happens close to the instrument and the dSCDs at different elevation angles can present similar values (i.e., their information content are no longer representative of different layers of the atmosphere). Thus these measurements are not very suitable for vertical profile retrieval.

P5 L32: What are typical DOFs of the data set?

A summary of the degrees of freedom is now provided in the Supplementary Material (Table S1). The reader is now kindly referred to it on P6 L1.

P5 L39: What is meant by 'lower differences'? By definition, the a-priori should be independent information (e.g. from a climatology). This sounds like the O4 dSCDs were used to optimise the a-priori scale heights prior to using them as information independent of the measurements. What could be an explanation for the difference and the very high SH of 2km?

As remarked by the referee, an ideal a priori should be independent information derived from observations. Unfortunately, observations in the Antarctic atmosphere are not abundant (this is now remarked on P6 L11:

*"As a consequence of its logistically-complicated location and to very harsh weather conditions, there are very scare observational data describing the atmosphere at Antarctica".*

Regarding our study on the vertical scale, besides the ozone-sondes performed at Belgrano, no other ancillary data are available. The ozone-sonde data have been used to model the pressure, density, temperature and ozone profile of the atmosphere at this station. Thus, we used a usual exponential decreasing function for both a priori profiles (extinction and BrO) and then we performed test varying parameters such as e.g. the correlation length, AOD, surface albedo, etc (just 2 or 3 different values) characterizing the a priori profile and its covariance matrix. Then, we compared the measured and resulting simulated dSCDs (O4 and BrO) to find the values providing the best agreement between both kinds of dSCDs (note that the O4 vertical profile is known in the atmosphere). In fact, this is pretty much the only test we can perform to check if the chosen parameters (a priori profile and covariance) allow to obtain reasonable values of the retrieved profiles.

P6 L36: As mentioned in section 2.2.1...: This is not discussed in 2.2.1 at all (but should be - based on averaging kernels)

"As mentioned in Sect. 2.2.1" is corrected to "As mentioned in Sect. 2.2.2." (referring to P5, L19-20 of the initial draft). See also our responses to General Comments.

P7 L12: Please make clear which station is referred to. (There is no Polar night at Marambio!) What is meant by 'BrO levels were undetectable just before...and immediately after... '? The data set provided here reports no MAX-DOAS observations within one month from both of these points in time. Does that mean no data or no BrO observations? (compare Fig. 3 and 4)

We agree with the referee, unlike at Belgrano, at Marambio there is no complete darkness as already shown by the hours of light at the station (in yellow in Fig. 4 of the draft). However, as stated in the draft when referring to the DOAS data presented, only DOAS data with SZA < 75° are considered in the work (P4 L40). This is now clarified in the new draft (P7 26-27, P9 L1, P10 L19, L26, L34, and in Figures 3, 4, 6, 7, 10 and 12-14).

P7 L13: What is meant by 'the magnitude and variability of the BrO maximums direct the difference'? What is the maximum referred to? A daily maximum?

*"VCD2km"* is now added on P7 L28 and L31 and "an absolute" is now added on P7 L29. Thus the sentences result as *"…, the magnitude and variability of the BrO VCD2km maximums direct the difference between both stations, with an absolute maximum BrO VCD2km observed in Marambio being 3.2 times higher than in Belgrano. As can be observed in Fig. 4, it is also worth noticing the clear photolytic activation of BrO in Marambio during austral spring, with levels an order of magnitude higher than the median BrO $VCD_{2km}$ values at the station…."*

P8 L4: the VMRs were not 'measured' rather estimated.. The claim of sensitivity up to 6km altitude should be substantiated (see General comments).

In the new draft *"BrO vmr measured"* is changed to *"the BrO vmr retrieved"* (P8 L20) and, accordingly, in the caption of Fig. 10 *"observed"* is changed to *"retrieved"*. Also, "the first 6 kilometres" (P8 L4 of the initial manuscript) is changed to "the lowest kilometers" (P8 L20). See also our responses to the General Comments.

P8 L5: Absence of elevated layers: If this actually can be inferred should be reviewed after assessment of the vertical sensitivity (see General comments)

The sentence (P8 L5-8 of initial draft)

*"One characteristic of these BrO observations is that, unlike conclusions of previous Antarctic studies suggesting the presence of reactive bromine above 4 km altitude (e.g., Frieß et al., 2004; Roscoe et al., 2014), during 2015 no elevated plumes of BrO were observed at either of the two Antarctic stations referred to in this work. In fact, during 2015, most of the BrO was confined within the first 2 km of the troposphere (Fig. 8)."*

has now been rephrased and moved to (P9 L19-24):

*"In previous studies of the Antarctic troposphere (e.g., Frieß et al. 2004 and Roscoe et al. 2014), the presence of uplifted reactive bromine was suggested. While the work of Frieß et al. (2004) remarked the presence of reactive bromine uplifted (> 4 km altitude) due to advection processes, in the work of Roscoe et al. (2014), the authors where able to differentiate between 2 types of BrO vertical profiles: those with only near-surface BrO (centered at 200 m) and those with double peak (centered at 2 km and near-surface). In the work here presented, no uplifted layers (> 2 km) of BrO were detected although, given the limited vertical information content of the MAX-DOAS observations (Supplementary Material), double peak type of BrO profiles cannot be ruled out."*

Please, refer also to our responses to the General comments.

P8 L23: Discussion of BrO vmr vertical profiles. These profiles should be generated based on information content/the analysis of the averaging kernels (see General comments). The presentation chosen here -without error bars and with a smoothed profile rather than visualising the single layers of the retrieval is misleading for readers not familiar with the details of MAX-DOAS profile inversions. In such a figure, the (typical) a-priori and its errors should be presented as well.

The a priori BrO profiles (and errors) are now included in Fig. 10 (no smoothing) and its caption. Note that the altitude scale now goes up to 3 km. See also our responses to the General Comments.

P8 L41: Whether or not the detection of BrO above 2km can be discussed with the presented data set strongly depends on vertical sensitivity.

Please, see our responses to the General comments.

P8 L41, P9 L1: The inversion error was not clearly defined (see comment p.5 l.25)

As mentioned on our response to comment on P5 L26, this is now clarified in the new draft (P5 L33-36).

P9 L3: BrO in the free troposphere: The absence of BrO and hence an upper limit for BrO in the free troposphere should only be concluded for altitudes where the MAX-DOAS observations are sensitive (based on averaging kernels, see general comments). Based on the data presented, in my opinion, the absence of BrO (above the stated detection limit) in the free troposphere can only be concluded, if one assumes an upper limit of the boundary layer of 1500m or higher because the retrieval quite consistently seems to yield mixing ratios of BrO of at least 1.5 pmol/mol above 1000m altitude (Figure 8 or Belgrano example from 29th October in figure 9). I am not convinced that an altitude of 2000m for the top of the boundary layer is a very regular event in polar regions. Indeed, the two cited publications show e.g. a maximum depth of the convective boundary layer of about 300m for Halley station (King et al. 2006) and give altitudes for the humidity inversion at Marambio of 700m-1300m throughout the year with values of 1000m in spring (Nygard et al 2013). The data as presented here, shows considerable mixing ratios above the detection limit between 700 and 1300m. If the vertical sensitivity at these altitudes is sufficient, discussions about the export of BrO to the free troposphere or the absence thereof should be based on information about the typical depth of the boundary layer at the respective times of the year at the two locations (e.g. the radiosonde data at Belgrano used in the retrieval or -if available the radiosondes at Marambio used in Nygard et al. 2013) rather than on the stated (quite wide) range of possible altitudes for the top of the boundary layer of 100m to 2km.

As mentioned on the initial manuscript (P9 L5-8), although the definition of the top of the boundary layer is out of the scope of our work, indeed a threshold between the boundary layer and the free atmosphere needs to be set in order to assess the detection of BrO in the free troposphere. Indeed, (P8 L4-5 of initial draft) *"former studies place the top of the boundary layer in Antarctica between 100 m and 2 km, depending on the boundary layer parametrization and time of the year (e.g., King et al., 2006; Nygård et al., 2013)"*. The definition of the top of the boundary layer is quite a topic by itself and it is usually based on the vertical structure of e.g. its chemical components, temperature, humidity, wind, turbulences, etc. Based on the 10-year climatology of Nygård et al. (2013) investigating the different humidity inversions above given sites (including Marambio but not at Belgrano), the (median) based height of the inversion at Antarctica was located between 0.9 km and 2 km during the non-winter periods (i.e., periods covered in this work). In particular, at Marambio the top of the boundary layer (based on humidity) was at an altitude between 1 and 1.4 km for these periods. Regarding Belgrano, based on monthly averaged temperature profiles obtained from the ozone-sondes at the site, the altitude of the inversion layer (if present) at Belgrano ranged from 1 to 1.5 km (depending on the month), while the lapse rate most of the months suggested a mixing layer height of up to 1 to 2.5 km (depending on the month). Although these data suggest that the top of the boundary layer could vary between 0.9 and 2.5 km depending on the station and time of the year, as the referee remarks, with the data that we have we are not able to define the altitude of the boundary layer at the specific time and dates of our observations. This is now made clear by rephrasing P9 L3-8 of the initial draft

*"…the BrO detection limit here provided may be regarded as an upper limit of BrO in the free troposphere since former studies place the top of the boundary layer in Antarctica between 100 m and 2 km, depending on the boundary layer parametrization and time of the year (e.g., King et al., 2006; Nygård et al., 2013). This upper limit of BrO in the free troposphere of Antarctica is consistent with the few previous studies of the vertical distribution of this trace gas in the Arctic and Antarctic regions that set upper limits of BrO in the polar free troposphere of 1.5 and 2 pmol mol-1, respectively (e.g., Frieß et al., 2011; Prados-Roman et al., 2011; Peterson et al., 2017; Hüneke et al., 2017)."*

to:

*"…former studies place the top of the boundary layer in Antarctica between 100 m and 2 km, depending on the boundary layer parameterization, station and time of the year (e.g., King et al., 2006; Nygård et al., 2013). Although, Nygård et al., (2013) marked the height of the boundary layer at Marambio between 1 and 1.4 km for the non-winter periods referred to in this work, and the ozone-sondes performed at Belgrano set that height between 1 and 2.5 km at that site (e.g., Parrondo et al., 2014). However, the altitude threshold between the boundary layer and the free troposphere cannot be assessed on the time resolution of our observations. Additional investigations at the two sites would be needed to confirm whether BrO reaches the free troposphere and, given the case, to assess the budget of BrO in the Antarctic free troposphere. Note that previous work on polar environments set BrO below 1.5 and 2 pmol mol$^{-1}$ in the Arctic Arctic and Antarctic free troposphere, respectively (e.g., Frieß et al., 2011; Prados-Roman et al., 2011; Peterson et al., 2017; Hüneke et al., 2017)"* (P9 L25-34).

*Accordingly, "In fact, in line with previous polar studies, this work sets an upper limit of BrO in the free troposphere of Antarctica of 1 pmol mol-1."* (P11 L14-15 of the initial draft) is deleted.
See also our responses to the General Comments with regard the altitude sensitivity of the DOAS data presented.
Worth mentioning is the fact that, in Sect. Sect. 3.1.1., the BrO confined within the first 2 km of the atmosphere (i.e., BrO VCD$_{2km}$) is the parameter used for comparing the budget of BrO above both stations. Also, the BrOx investigation is based on surface studies. Therefore, our conclusions to these regards are independent of the location of the top of the boundary layer.

P9 L24: Estimation of BrOx: Please make the assumptions going into this clearer. I do not understand at all how the observational data feeds into this estimation. Is this an estimation at noon? Was the daily average of BrO and O3 used or the maximum? The rate of BrO photolysis assumed here should be stated. This rate strongly depends on the actinic flux which in turn depends on visibility conditions/cloud cover. How was this treated exactly? What influence does the filtering of total AODs above 5 mean for this? HO2 has a very pronounced daily cycle as well. What justifies the selected, fixed mixing ratio?

Thank you for noticing that we forgot to write down the value considered for the photolysis rate coefficient of BrO. That value is now stated on P10 L10.
Regarding the BrOx estimation (Eq. (2)), as already mentioned on P10 L13 of the initial draft, the calculations presented are a *"simplified scheme"* and are *"based on conclusions after numerical models and laboratory and campaigned-based observations obtained in the polar regions (mainly the Arctic)"* (P10 L25 of initial draft). We do consider that, as already stated (P10 L38-40 of initial draft), there is a *"need of further investigations for a better understanding of all the processes and key parameters involved in the halogens' pathways in the Antarctic troposphere"* (in the new draft the word *"long-term"* is now added to that sentence, P11 L31). Hence, given the lack of Antarctic literature of many (most) of the required parameters involved in the bromine activation, in Eq. (2) we have used fixed mean values found in literature for polar regions (even if those were from the Arctic region) for all the parameters except for [BrO] and [O3] (where we have used the observations, as already stated on P9 L21-22 of the initial draft). As for the [HO2] (and [OH]) mentioned by Referee #1, the fixed value used (P9 L20 of initial draft) corresponds to the mean concentration of HO2 (and OH) as observed at the Antarctic station of Halley (e.g., Bloss et al., 2007). In the case of [ClO], the value used (P9 L20 of initial draft) is an average value for Arctic conditions (e.g., Halfacre et al., 2014). We agree with the referee and these sort of details are relevant and should be stated in the text. We have clarified now those details in the new draft (P10 L42).
Worth mentioning is that we consider that (P10 L11-13 of initial draft) *"Additional investigations on the variability and geographical distribution of the bromine source gases throughout the year are suggested to address the bromine pathways in the Antarctic troposphere and their consequences"*. Note that, on P10 L36-39 of the initial draft, we conclude that the kinetic calculations based on Eq (2) are close to reality (i.e., to observations) in the case of Marambio

but not in the case of Belgrano. Also, as stated on P11 L31-34 of the initial draft, *"…dedicated investigations combining collocated observations of e.g. halogenated substances (not only BrO), organic compounds, DMS, NOx, HOx, particles and sea ice properties at different stations, could assist a thorough study of the bromine sources and pathways at Antarctica, their geographical distribution and their projections under a changing environment"*. Note that *"models and"* is now added to that sentence (P12 L24) to reinforce the need of model-observation synergies.

P9 L31: What exactly is shown in this figure? Daily averages? maximum values?
The figure shows BrOx (Br + BrO) based on Eq. (2) where [O3] and [BrO] are the observed near-surface BrO and O3 under low wind conditions for the timestamp of the MAX-DOAS measurements. This is now clarified in the draft (P10 L17). Throughout the draft, "near surface" has been changed to "near-surface" (P3 L19, P6 L13, P7 L34 and P8 L6).

P9 L40: What is meant by 'troposphere reactivity'? oxidation capacity?
"tropospheric reactivity" is now changed to "oxidizing capacity" (P10 L28)

P10 L13: This seems a bit circular to me. The O3 measurements were used to calculate BrOx which then is used to estimate the O3 loss rate? Please elaborate your reasoning behind this.
As mentioned throughout the draft (e.g., P3 L8-14 of initial draft), there are very few observational data in the Antarctic environment which forces us to make several assumptions. In this sense, as mentioned above and on P9 L18-41 and P10 L1-11, the BrOx is calculated through Eq. (2) by fixing all the parameters (to values observed in polar regions) except for BrO and O3 (which are the observations).
Later on (P10 L13-24 of the initial draft), those BrO and O3 observations are also used to estimate the lifetime of ozone at each site based on Eq. (1), i.e., assuming that the main O3 depletion is through the bromine-chlorine channel (P10 L37-38 of initial draft) and that the [ClO] is fixed as in Eq. (2) to average Arctic conditions (the later is now included in the new draft, P10 L42). To clarify this even further, *"BrOx regimes"* is now changed to *"BrO and O3 regimes"* (P11 L1).
Since we are aware that these are very simplified approximations and could not be realistic, we do compare (P10 L25-40 of initial draft) *"these observed τO3 with the τO3 estimated above from kinetics"* for given days aiming at evaluating the validity of the simplifications made in our estimations. Note that we extract the *"observed τO3"* from the *"the ozone observations"* (P10 L29 of the initial draft), i.e., form the data shown in Fig. 7 of the initial draft. As stated on P10 L36-38 of the initial draft, *"The resemblance of the observed and calculated τO3 at Marambio suggests that the assumptions made at Marambio's surrounding (e.g., the Br-Cl channel dominates the ozone depletion) is close to reality which seems not to be the case for Belgrano's surroundings"*. Thus, through this exercise we investigate the goodness of our (very simplified) estimations and clearly state that, while the observations at Marambio suggest that the assumptions made at this site are not far from reality, those assumptions are limited in the case of Belgrano's surroundings. Since this observation vs. estimation exercise based on exemplary days have no statistical meaning (this statement is now included in the new draft P11 L28-29, we conclude that there is a *"need of further long-term investigations for a better understanding of all the processes and key parameters involved in the halogens' pathways in the Antarctic troposphere"* (P10 L38-40 of initial draft) *"combining collocated observations of e.g. halogenated substances (not only BrO), organic compounds, DMS, NOx, HOx, particles and sea ice properties at different stations"* (P11 L32-34 of initial draft). In this regard, we consider that the observation vs. estimation exercise presented is therefore justified and offers some further information of

our observations (e.g., at Marambio the dominant ozone chemical loss is driven by the Br-Cl channel while at Belgrano most probably there are more species involved).

P11 L4: I would add 'at two (new) sites' after 'inorganic bromine'. The sentence as it is sounds a bit as if the results apply to the entire Antarctic troposphere while the observations are already quite different for the to sites (as mentioned by the authors later on)
The sentence *"This study reports on the presence and distribution of reactive inorganic bromine (BrOx) in the Antarctic troposphere"* is now changed to *"Based on contemporary ground-based observations performed during 2015 at two Antarctic sites (new sites as far as tropospheric BrO observations is concerned), this study reports on the presence and vertical distribution of reactive inorganic bromine in the low troposphere at the two sites and discusses the geographical distribution of BrOx"* (P11 L35-38).

Table 1: measurement period column: Please only use months and days here. The use of the term 'season' in an Antartic context is misleading as it could also mean three (summer) seasons. The indicated periods from this study also should rather be 4.5 months than '3 seasons' since the periods from the other publications also only state periods of reported observations and not the entire time when the instruments were deployed.
The periods of the works of Kreher et al. (1997), Saiz-Lopez et al. (2007) and this work are clarified in the new draft (Table 1).

Table 3: days with snowfall: Is this only precipitation (excluding blowing snow)? Information about days with blowing snow would be interesting as well - if available.
The snow fall provided in Table 3 is taken from the 2015 statistics offered by the WMO at each station and are based on surface synoptic observations. This is now clarified in the header of the table. Unfortunately there are no statistics on blowing snow conditions as such. Instead, we can use the wind speed observations threshold of 12 m s$^{-1}$ (e.g., Jones et al. 2009) for blowing snow. In the new draft we have excluded the Fig. 5 (suggested also by Referee #2) and hence the other figures re-numbered. Also, as mentioned on our responses to the General Comments (1.2), we have modified the figure with the wind roses (former Fig. 6) in order to include statistics concerning the wind speed and directions as suggested.

Figures3+4: Please provide errors and detection limits in these plots
For sake of simplicity of the plots, the relative errors of the AOD$_{2km}$ and BrO VCD$_{2km}$ are now provided in the draft (P7 L12 and P7 L25, respectively)

Figure 5: This data is not very helpful. The averaging window could be increased to show wind regimes in different seasons. Alternatively, a histogram of wind speeds would provide more information.
Figure deleted in the new draft (see our response to Table 3).

Figure 6 (now Figure 5): Additionally, data filtered for the days presented in the profile data would be desirable
This is now shown by the color code (see our response to Table 3).

Figure 7 (now Figure 6): It would be nice to have the reported periods of MAX-DOAS observations marked in these plots.
The periods with MAX-DOAS observations are now indicated (hence the caption adapted).

Figure 8 (now Figure 7): These plots are quite small. A blow up or separation of the two periods would benefit the information conveyed by them. The vertical axis should be adapted to the updated vertical sensitivity. Plotting vmr boxes with the size of the retrieval grid (pixels) or aggregated profiles based on averaging kernels rather than using the linear(?) smoothing in the

colour map plots would make the nature of the retrieval process and the resulting data clearer. It would also be good to clearly mark periods where data is missing or was filtered out (e.g. based on the AOD limit). For example, it is not clear to me if the periods in the BrO profiles from Marambio in December are missing data or just no BrO at all for half a month.

We appreciate for pointing out these details so we can improve the color plots since, as remarked, they do contain a lot of information. The vertical scales have now been adapted (see our responses to General Comments). Also observations below detection limits are now shown in black and the times when the data are not above the quality filters are now indicated in gray boxes (this is now mentioned in the caption). Indeed, these sort of (often used) contour plots interpolate linearly across rectangles (vertical grid and time stamp). However, we do consider that no relevant information is lost by using these plots. In addition, by showing the plots in this manner (the whole measurement period at each station for AEC and for BrO) it is easier to gain an overview of the measurements at both sites and to gain an insight on the message contained on each plot (e.g., altitude range, temporal range, AEC or BrO variability in time and altitude, periods with no observations due to high SZA, etc.).

Figure 9 (now Figure 8): See remarks to Figure 8 regarding smoothing and axis. For the example day from Marambio on September 25th, the data selection should be reviewed. On that day, the SZA limit stated on page 4 of 75 degrees is reached already at 19:25 while the plot shows data until 20:00.

Please, refer to our response above. Regarding the timestamp vs. SZA issue the referee realized, unfortunately when drafting the plots of Marambio 25th Sept and Belgrano 11th November, there was a mistake on the decimal day to time conversion for the axes. All the timestamps have been checked for all the plots. Thanks for noticing.

Figure 10: Dots to indicate the values in the different levels of the retrieval, error bars and a-priori profiles with the respective errors should be added here

The a priori and error has now been included and no smoothing is applied in the plots. Also, the vertical grid (100 m) is now shown with small ticks (indicated now in the caption).

Figure 11: The location of Belgrano and Marambio is the only new information in this figure. Maximum values are already provided in table 1. As these values are from different years, plotting them in such a manner could be misleading.

In addition to the location of Belgrano and Marambio as new sites regarding ground-based tropospheric BrO observations, this figure also contains information regarding the geographical distribution of BrO. Indeed, as it appears in the caption of the figure *"only the present study provides contemporary observations from different sites"* (i.e., Marambio and Belgrano- making this work particularly relevant). Moreover, through this figure, the reader can easily appreciate the strong difference on BrO between the two new sites and, also, have an overview of the up-to-date maximum values of BrO on the (few) other sites in Antarctica. Even if those other works refer to different years, they also cover the bromine active period and this plot can *"point(ing) once more its heterogeneity with regard to reactive bromine load"* (P8 L38-40 of the initial draft). Thus, we consider this plot is worth keeping.

Figure 12: What is meant by 'depicts the BrOx [...] at each station' Please make clear what is plotted here. Daily averages or maximum values?

The figure shows the BrOx (i.e., BrO + Br), where BrO derives from the observations and Br from Eq. (2). Since that equation depends on O3 (which is also an observation), this parameter is used as color code. The time stamp is that of the BrO measurement. Please refer also to our response to the specific comment to P9 L24.

**Technical comments**

P2 L 27: added

P4 L3: removed

P4 L31: changed

P5 L14: we consider the referee refers to P4 L14 (changed)

P5 L11: changed

P5 L32: corrected

P6 L19: corrected

P6 L32: added

P8 L33: corrected

P10 L13: corrected

P10 L25 (now P10 L30): corrected

P11 L9 (now P11 L14): corrected

Table 1: corrected

---

## Author Comment (AC2) · 24 May 2018

The authors are grateful for Referee #2's time and constructive comments and the appreciation of our work. In the following, we address his/her comments. Please, note that specific details on the MAX-DOAS and inversion technique has now been included on a Supplementary Material.

The Referee Comments are in black and Authors Comments in blue. References to pages and lines of the draft are indicated with "P" and line "L", respectively.

**Specific comments**

MAX-DOAS analysis

The authors claim that BrO is not present in significant amounts above 2 km based on their ground-based measurements, where they retrieve BrO vertical profiles from 0-6 km. I am skeptical that ground-based MAX-DOAS measurements can be used to make this claim. For what it is worth, I am skeptical that the prior studies cited could actually observe BrO at those altitudes as well. The information content outside of the lowest elevation angle measurements simply isn't high enough. The authors should present averaging kernels showing that the measurements are sensitive to changes in BrO above 2 km if they are going to make this claim. I also think the presented vertical profiles should also be limited to 2 km unless the averaging kernels show that a higher altitude is merited.

Tests performed with the ratiative transfer confirm the sensitivity of the technique to up to 4 km and 2.5-3 km (aerosols and BrO, respectively). We kindly ask the referee to see the Supplementary Material (Sect. 3) and also our responses to the General Comments 1.1. of Referee #1.

Sea ice conditions between the two sites

I think the author's points about needing to examine the sea ice conditions at both sites and the heterogeneity being potentially linked to sea ice differences is a good one. However, I think simply describing the sea ice around Marambio as seasonal without providing further detail is potentially misleading, as the ice toward the outside edge of the sea ice in the Antarctic is often the oldest sea ice (excluding the "permanently" sea iced sections surrounding Belgrano) (Nghiem et al., 2016). This older sea ice is likely lower salinity than the newer sea ice regions closer to the coast. These differences may impact the overlying snowpack, which is the likely source of the reactive bromine. Of course the proximity of this older ice to open water may also lead to enhanced snow salinity due to sea spray aerosol deposition (e.g. May et al., 2016). In any case, I'd like to see the authors add a more detailed discussion of the sea ice conditions at the two sites.

Indeed, although the mentioned work of Nghiem et al. (2016) states that the "Antarctic sea ice is dominated primarily by first-year sea ice" which is often linked to bromine explosions (e.g., Simpson et al., 2007), dedicated investigations into the link between the properties of the sea ice and the composition of the tropospheric in Antarctica are definitively worth considering (although not easy to undertake from the logistic point of view...).

With regard the data we present in this work, we have now added a new figure (Fig. 9) with closed-up views of the sea ice conditions nearby Belgrano and Marambio on the selected days of Fig. 8 (from end of September to end of November). Note that this new figure does not substitute Fig. 1 since the later provides the reader with a quick overview of the locations of both stations within Antarctica (something missing in this new zoomed in figure). Note that, as stated on the caption of the new figure, "in barely 1 month (25th September - 29th October) the sea ice surrounding Marambio underwent strong transformation, going from medium/highly concentrated sea ice in September and with barely permanent open waters (upper left figure), to pretty much complete open ocean (disappearing all the sea ice beyond 50° W). During the timeframe of that sea ice transformation, BrO VCD$_{2km}$ peaked at Marambio (Fig. 4). Also, note how the edge of the sea ice nearby Belgrano transforms towards November (e.g., lower right).".

Page1, Line 41: This sentence should have references for these impacts of atmospheric halogens.
As appearing in P2 L4 of the initial draft, the reader is kindly directed to the compendium work of Simpson et al. 2015 on "Tropospheric Halogen Chemistry: Sources, Cycling and Impacts" and the studies referred to in it.

P5 L32: What percentage of the retrievals has a DOF larger than 1?
Details on the DOF are now provided in the Supplementary Material (and indicated on P6 L1 of the new draft).

P5 L40: A summary of the degrees of freedom for the BrO retrievals should be presented here as well.
It is now included in the Supplementary Material.

P6 L39: Can you state the differences in AOD between the two sites more quantitatively?
In the new draft (P7 L14-15) now we add "At Belgrano, 62 % of the $AOD_{2km}$ was lower than 0.05 (12 % between 0.05 and 0.1) compared to 90 % of $AOD_{2km}$ at Marambio that was below 0.05". For clarity, the subscript "2km" has been added to AOD in the vertical axes of Fig. 3 (i.e., $AOD_{2km}$). Please note that, as mentioned on our draft (P7 L7 of the initial draft), a forthcoming publication will focus on the aerosols observed at the two stations (Gómez-Martín et al. 2018).

P7 L11: $0.8 \times 10^{13}$ molec $cm^{-2}$ isn't a range as presented. Please clarify, is this a standard deviation?
"range" is changed to "value" on P7 L25 (i.e., 75% of the data BrO VCD2km were < $0.8 \cdot 10^{13}$ molec $cm^{-3}$)

Suggested figure modifications:

1. Figure 3, 4: I don't really think it is necessary to shade regions without data. It gives the figure a cluttered look.
We appreciate the referee suggestion. However, we do consider that by including the "shades" the reader can have an overview of the amount of data contained in this work at two different sites. Note that by excluding the shaded regions the reader could interpret that we did have MAX-DOAS data throughout the year. We would like to clearly state (not only in the text) that there were periods of time with no MAX-DOAS data (either due to instrumental issues or to SZA > 75°, this now clarified in the caption of both figures and throughout the draft (P7 26-27, P9 L1, P10 L19, L26, L34, and in Figures 3, 4, 6, 7, 10 and 12-14).

2. I think just showing Fig. 6 is sufficient, and the timeseries of wind speed (Fig. 5) isn't really needed.
We agree. In the new draft the Fig. 5 is deleted and the wind roses plots binned by wind speeds so the information is not lots. Please, see our responses to the General Comments (1.2) of Referee #1

3. Fig. 7: Consider plotting both ozone series on the same panel so one can clearly see the differences between the two sites.
Similarly to most of other figures of the work where the information is differentiated between stations (i.e., one plot per station), we'd rather plotting the time series of ozone also in a two panel manner. Moreover, it would be quite complicated to distinguished features when plotting the O3 time series in one single panel (note the noisy data of Marambio, for instance). Also,

please bear in mind that we have now added the time periods with MAX-DOAS observations at each station (making the "one-plot" way even more complicated to understand).

4. Fig. 8,9,10: As I suggest above, the portion above 2 km should be cut unless you can demonstrate that your retrieval is sensitive to the true atmospheric state at higher altitudes.

Referee #2 is kindly referred to the Supplementary Material and to our responses to the General Comments (1.2) of Referee #1.

---

## Author Comment (AC3) · 24 May 2018

**Supplementary Material**

Herein we provide further details regarding:

1. Statistics of the DOAS fits
2. Retrieval of the vertical profiles after the MAX-DOAS observations
3. Tests to investigate the vertical sensitivity of the MAX-DOAS observations

**1. DOAS fit statistics**

The statistic of the retrieval of the BrO differential Slant Column Density (dSCD) and its errors given to the inversion scheme are shown in Fig. S1. Overall, at both sites the median BrO dSCD decreased with the elevation angle of the scan (median relative error between 21 - 39 %).

[Figure]

**Figure S1: Box chart of the BrO dSCD (left figure) and percentage relative error of the BrO dSCD (right figures) retrieved from Belgrano and from Marambio data (upper and lower figures, respectively).** The vertical scale indicates the elevation angle of the retrieved dSCD (depending on the station), while the horizontal scale indicates the range. In all the figures, the boxes provide the 25th to 75th percentile while the median values of the data set are indicated with dashed lines.

In the case of O4, the statistics of the DOAS fit are shown in Fig. S2. Overall, at both sites the median $O_4$ dSCD decreased with the elevation angle of the scan (median relative error 1.5 - 6 %).

[Figure]

**Figure S2: Box chart of the $O_4$ dSCD (left figure) and percentage relative error of the $O_4$ dSCD (right figures) retrieved from Belgrano and from Marambio data (upper and lower figures, respectively).** The vertical scale indicates the elevation angle of the retrieved dSCD (depending on the station), while the horizontal scale indicates the range. In all the figures, the boxes provide the $25^{th}$ to $75^{th}$ percentile while the median values of the data set are indicated with dashed lines.

**2. Retrieval of vertical profiles after the MAX-DOAS observations**

Degrees of freedom:

In Table S1 below, the reader can find a summary of the degrees of freedom (DOF) of our retrievals. Overall, 93 % and 97 % of the aerosol retrieval at Belgrano and Marambio (respectively) presented DOF > 1. Regarding the retrievals of BrO profiles, 65% and 71% of the retrievals at Belgrano and Marambio (respectively) had DOF > 1.

**Table S1: Summary of the degrees of freedom of the profile retrievals performed throughout this study.** Mean and standard deviation of the degrees of freedom for the inversion of aerosol extinction and BrO vmr at both sites.

| Research site | Inversion | Mean | Standard Deviation |
|---|---|---|---|
| Belgrano | Aerosol | 2.21 | 0.99 |
| | BrO | 1.16 | 0.13 |
| Marambio | Aerosol | 1.58 | 0.74 |
| | BrO | 1.21 | 0.12 |

Examples of retrieved Averaging Kernels and profiles

5   Figures S2 and S4 show examples of the averaging kernels (AK) of the aerosol and BrO profile retrievals, respectively. Figures S3 and S5 show the inverted profiles of the aerosols and BrO (respectively) corresponding to those AK.

Overall, most of the information content of our DOAS measurements was higher below an altitude of 1.5-2 km. However, there were days where the particular extinction conditions of the atmosphere (i.e.,
10   scattering and/or absorption) rendered information content not negligible up to 4 km (e.g., Fig S2, right). Please, refer to Sect. 3 of this Supplementary Material, for investigations on the vertical sensitivity of the MAX-DOAS method given (realistic) extinction conditions different than those occurred during our measurement period of 2015.

15

[Figure]

**Figure S2: Example of averaging kernels corresponding to extinction profile retrievals at Belgrano.** The averaging kernels correspond to the observations performed at Belgrano on (left) 7[th] October 2015 (12:28 UTC) and (right) 30[th] November 2015 (09:07 UTC). Note how in the right figure, the sensitivity towards elevated layers
20   is not negligible compared to the left figure.

[Figure]

**Figure S3: Example of aerosol extinction profiles retrieved at Belgrano.** The retrieved profiles correspond to the averaging kernels shown in Fig. S8 (left: 7[th] October 2015 at 12:28 UTC, right: 30[th] November 2015 at 09:07 UTC). The blue lines indicate the retrieved BrO vmr profiles (blue shaded areas indicate its error) and the a priori AEC profile is shown in cyan in both figures.

[Figure]

**Figure S4: Example of averaging kernels corresponding to BrO profile retrievals at Belgrano.** The averaging kernels correspond to the observations performed at Belgrano on (left) 7[th] October 2015 (12:28 UTC) and (right) 1[st] December 2015 (08:42 UTC). Note that, on the right figure, the contribution of layers above 1km to the retrieval of BrO at the surface is negligible (while this is not the case on the left figure).

10

15

[Figure]

[Figure]

**Figure S5: Example of BrO mixing ratio profiles retrieved at Belgrano.** The retrieved profiles correspond to the averaging kernels shown in Fig. S10 (left: 7th October 2015 at 12:28 UTC, right: 1st December 2015 at 08:42. UTC). The red lines indicate the retrieved BrO vmr profiles (red shaded areas indicate its error) and the a priori BrO profile is shown in cyan in both figures.

**3. Tests to investigate the vertical sensitivity of the MAX-DOAS observations**

As mentioned above, the vertical sensitivity of the MAX-DOAS observations depends on the extinction conditions of the atmosphere and the most probable altitude of scattering which also depends on the viewing geometry. Thus, we have performed sensitivity tests with our raditative transfer model (RTM) and inversion scheme to simulate extinction conditions that, although realistic, did not happen in the time frame of our measurements. With these tests, we will investigate the vertical sensitivity that our MAX-DOAS observations are able to achieve. In this sense, we have simulated different aerosol scenarios with layers of AEC = 0.15 km$^{-1}$ located at different altitudes (peaking at 2, 3, 4 and 5 km, see Fig. S6) and include them in our RTM taken the parameterization of the atmosphere at Belgrano as the test bench (i.e., the elevation angles of the MAX-DOAS observations at Belgrano and the same vertical grid, ground albedo, P, T, geometry, ect. used for Belgrano on 11th November).

20

[Figure]

25

**Figure S6: Different aerosol extinction vertical profiles used for investigating the vertical sensitivity of the MAX-DOAS observations.** The different plain shaded areas indicate the aerosol layer centered at different altitudes in the parameterization of the atmosphere for the different tests. As an example, the inverted profile for the 10:40 UTC scan (Belgrano, 11th November) is provided in dashed green (i.e., smoothed true profile).

The forward modelling of the O$_4$ dSCDs with different aerosol scenarios (from no aerosols to the aerosol scenarios shown in Fig. S6) is provided in Fig. S7. In this figure, one can see that, given the elevation angle of the observations at Belgrano, our observations wouldn't be able to distinguish between the aerosol layer centered at 2 km (red line) and one centered at 3 km (in blue), at least not beyond the measurement error. However, observations should be able to distinguish between an aerosol layer at 4 km and an aerosol layer at 5 km. Moreover, observations should be able to capture the difference between an aerosol layer located at 2 km and one located at 4 or 5 km (see for instance the modelled O4 dSCDs at the elevation angle of 30º, blue and green).

[Figure]

**Figure S7: O$_4$ dSCD under the aerosol scenarios shown in Fig. S4.** The different colors refer to the forward modelled dSCD under the different aerosol scenarios shown in Fig. S6 (Belgrano, 11[th] November, 10:14 UTC). Additionally, the measured dSCD at that moment are also sown in bright green.

Also, in order to investigate the differences between a true uplifted aerosol profile and the retrieved one (i.e., the smooth version of the true profile), using the same a priori aerosol extinction (AEC) profile and covariance as the one used throughout our work, we can retrieve the aerosol profiles from the modelled O4 dSCDs with an aerosol layer peaking at 2 km and also from the modelled O4 dSCDs with an aerosol layer peaking at 4 km. The resulting retrieved AEC profiles are provided in Fig. S8. Note that, although the retrieved aerosol profiles peak at slightly lower latitude than the true profile, there is a clear difference between the altitudes of the maximum AEC retrieved at both instances. In fact, when investigating the averaging kernels corresponding to the aerosol inversion from the modelled O4 dSCDs with an aerosol layer peaking at 4 km (Fig. S9), the retrievals at 3.5 and 4 km are sensible mainly to layers above 2 km. This indicates that our MAX-DOAS observations and profile inversion scheme is able to sense the difference between a layer below 2 km and one at 4 km. Although these simulations indicate that the method is limited to quantitatively retrieved layers aloft, it also shows that, in case of uplifted layers, the retrieved profiles would show the enhancements of AEC at higher altitudes than the ones that we observed when aerosols are close to the surface.

Similarly, sensitivity tests performed with a BrO layer uplifted at different altitudes indicate that our procedure would, for instance, be able to sense a BrO layer centered at 2.5 km. As shown in Fig. S10, in that case and despite the BrO a priori profile peaking at the surface, the retrieval would show a smoothed version of an uplifted BrO layer (at ~1 km) and also negative BrO vmr at the surface (with nonphysical meaning). Note that none of the profiles presented in this work showed this sort of behavior (i.e., no BrO layer above 2 km). Unlike the work of Roscoe et al. (2014), the vertical information content and sensitivity of our MAX-DOAS observations and inversion scheme would not be able to distinguish an uplifted layer of BrO if BrO is also present on the lowest layers of the atmosphere (i.e., in case of a double peak profile). In that "double peak" case, the sensitivity tests indicate that the BrO layer aloft would increase in 20 % the BrO column inferred below that altitude

(i.e., VCD2km) and ~10 % the mixing ratios retrieved just above the surface. Note, that the effect of such double peak on the retrieved BrO VCD2km and surface values would also depend on the scattering conditions of the atmosphere.

The sensitivity test performed support the definition of the vertical sensitivity of our MAX-DOAS observations throughout our work (up 4 km for aerosols and up to 2.5-3 km for BrO).

[Figure]

[Figure]

**Figure S8: Example of true aerosol extinction profiles vs. the retrieved one (i.e., smoothed true profile).** The left figure compares a true profile of an aerosol layer centered at 2 km (shaded in red) with the retrieved profile which peaks at ~1.5 km altitude (in black). The right figure compares a true profile of an aerosol layer centered at 4 km (shaded in pink) with the retrieved profile that peaks at ~3 km (in black). The a priori profile is provided also in both figures (cyan). Belgrano, 11th November, 10:14 UTC.

[Figure]

**Figure S9: Averaging kernels corresponding to the sensitivity test of inverting an aerosol layer at 4 km (Fig. S5, right figure).** Note how the retrieval at 3.5 and 4 km (orange, and grey, respectively) are sensible mainly to layers above 2 km.

[Figure]

**Figure S10: Sensitivity test of an uplifted BrO layer.** (Left) Comparison of a true profile of an uplifted BrO layer centered at 2.5 km (dark blue) with the retrieved one (i.e., smoothed true profile, in black). No aerosol load was included on the simulation. The a priori BrO profile used in the inversion is shown in cyan. The scan of the test corresponds to Belgrano, 11[th] November (14:52 UTC). (Right) Averaging kernel of the retrieval.